# Does Air Pollution Affect Health and Medical Insurance Cost in the Elderly: An Empirical Evidence from China

**Tianlei Pi [1,\*], Hongyan Wu [1] and Xiaotong Li [2]**

1  School of Economics and Business Administration, Chongqing University, Chongqing 400030, China; hongyanwu@cqu.edu.cn
2  Institute for Hospital Management, Tsinghua University, Shenzhen 518055, China; lxt17@mails.tsinghua.edu.cn
*  Correspondence: pitianlei@cqu.edu.cn

**Abstract:** In recent years, the health conditions of the elderly (the middle-aged and the old, which for this study means people over 50 years of age) have deteriorated along with the aggravation of air pollution, which led to the change of medical insurance costs. This phenomenon is particularly prominent in developing countries, such as China. A total of 15,892 research subjects from 56 prefecture-level cities in 23 provinces were collected from the database of China Health and Retirement Longitudinal Survey (CHARLS). We investigated the effects of air pollution, physical health, and medical insurance costs among three mechanisms using logistics and Ordinary Least Squares (OLS) hybrid cross-sectional regression, and we conducted a robust test. Overall, two pollutants, namely, PM10 and $NO_2$, respectively showed an "inverted U-shaped" and "positive U-shaped" influencing path to health. In addition, when we studied the mechanism of air pollution affecting medical insurance costs, we found that air pollution can affect medical insurance costs through affecting self-rated health, and that the impact path is related to different diseases to some extent. At the same time, there was a certain negative correlation between air pollution and medical insurance: The higher the degree of air pollution, the worse the self-rated health, and the fewer opportunities there are to purchase medical insurance. It can be seen that air pollution affects the physical health of middle-aged and elderly people, thus indirectly and negatively affecting the medical insurance cost. Further research also found that the types of air pollutants in southern and northern China showed some differences. Specifically, $NO_2$ and $SO_2$ were the pollutants that harm the health of the elderly in the south and north, respectively.

**Keywords:** air pollution; physical health; medical insurance cost; hypertension; heart disease

## 1. Introduction

In recent years, with the rapid development of emerging economies, such as in China, the world has experienced several problems including population explosion, over-consumption of resources, and ecological destruction, which have prevented economic development, as well as an improvement of life quality. Furthermore, these problems are threatening the sustainable development of humankind in the future. Among these problems of environmental pollution, air pollution accounts for a large proportion.

In fact, the impact of air pollution on human health has caused widespread concern. Globally, 3.3 million people die prematurely each year due to outdoor air pollution [1] Numerous studies have shown that air quality can affect public health [2–7]. Pope [2] was first to find that the negative effect of air pollution on public health would gradually increase over time. Grant and Poursafa [3] examined the relationship among air quality index (AQI), cardiac risk factors, and cancer mortality

in Iran and the United States. With the concern of air pollution from all communities, an increasing number of research has focused on the relationship between air pollution, health, and medical care. Li [4] analyzed the relationship between daily Air Pollution Index (API) and pollution indices, and between API and deaths using a distributed lag nonlinear model, which indicated that API can be used to inform people about the effects of air pollution on health in different populations. Geriatric people, females, and people with low educational attainment are the vulnerable groups affected by air pollution; so this information indicates who should be given more attention. Sun [5] collected data about residents' understanding and the increase of their medical expenses caused by air pollution through questionnaires. The author concluded that smog could directly affect the health status of residents, and increase their outpatient and emergency visits, as well as the incidence of chronic diseases. Xu [6] estimated the harmful effects of air pollutants, such as $SO_2$, on human health based on inter-provincial panel data. Liu [7] assessed the effects of air pollution on health using a semi-parametric regression model in four municipalities, and found that a nonlinear relationship exists between API and respiratory diseases, and that the impact of air pollution on health is highly significant. Hamedian [8] utilized a total of 11 parameters (i.e., $CO_2$, $SO_2$, PM10, PM2.5, $O_3$, $NO_2$, benzene, toluene, ethylbenzene, xylene, and 1, 3-butadiene) in a fuzzy inference system and a fuzzy C-means clustering method in order to evaluate five monitoring stations in Tehran, Iran. However, several types of air pollution indices have not been measured in China. Therefore, in this study, we used the concentrations of PM10, $SO_2$, and $NO_2$ to measure the degree of air pollution in the country.

Notably, although the economic growth in China has maintained its rapid growth since the 21st century, the problem of environmental pollution has become increasingly serious. As a result, widespread damage to human health has caused increasing concern. Particularly, air pollution and its health damage are prominent in the environment. According to the report on China's environmental status in 2015, up to 265 (78.4%) of the 338 prefecture-level cities exceed the standard of air quality. Current research has also shown that air pollution leads to respiratory and cardiovascular diseases, which increases the rate of morbidity, attendance, and mortality. Short-term exposure to air pollution can also increase the risk of respiratory and cardiovascular diseases in healthy populations. Poursafa and Rashidi [9] analyzed the relationship between the pollutant standards index and hematologic parameters, as well as respiratory diseases, in Iran. Pope [2] found that for each 10 μg/m$^3$ increase in the concentration of PM10, the risk of total deaths in the population increases by approximately 4.0%. Chen [10] found that reducing the concentration of PM2.5 can effectively reduce the loss of life expectancy. Hanne Krage Carlsen [11] used time series regression to investigate the association between the daily number of individuals who were dispensed anti-asthma medication and levels of the air pollutants particle matter. She found that there was a measurable effect on health, which could be attributed to the air pollution in Iceland.

Despite the theoretical and factual evidence that reveals that air pollution in China has become a major factor endangering people's physical and mental health, the existing literature on the impact of air pollution on health is mostly based on the investigation of a particular area, and the research on the true impact on micro-subjects (individuals) is lacking. In addition, the continuous rise in medical costs, including medical insurance, has become a global phenomenon. Maintaining a reasonable increase in medical expenses and coordinating the development of public health with the social economy has become the common goal of most countries [12]. Moreover, numerous analyses of consumption and investment needs on health, reflect that there is some economic impact on individuals and the society from physical health [13–18]. Some studies have examined the influencing factors and control strategies of medical expenses from the perspectives of supply and demand of medical services and prices. However, in developing countries where the current environmental pollution is becoming increasingly serious, such as in China, the existence of other conduction mechanisms for the significant increase in individual medical insurance costs due to the increase of air pollution remains unknown.

Therefore, based on the existing literature and the data from CHARLS of 2011, 2013, and 2015, this study attempts to systematically analyze the relationship between air pollution, physical health,

and medical expenses in recent years. China Health and Elderly Tracking Investigation Database adopts a multistage sampling method. In the sampling stages of counties, districts, and village houses, a probability sampling method is used according to population scale. A total of 150 districts and 450 villages were randomly selected from the 30 provincial-level administrative units in China. After excluding the cities with missing data, 15,892 sample units were obtained from 56 prefecture-level cities in China, covering 23 provinces (e.g., Hebei, Zhejiang, Yunnan, and Jiangxi) and four municipalities (e.g., Beijing, Tianjin, Shanghai, and Chongqing), as well two autonomous regions (i.e., Guangxi and Ningxia), which cover nearly 80% of China's provincial units and reflects the current situation in the country.

We used the self-rated health, outpatient, and inpatient of micro-subjects as the basic measure of the health condition of an individual. China has only started the statistics of the current widely used indicator, PM2.5, in 2013; thus, to ensure the consistency of statistical indicators before and after 2013, we substituted PM2.5 with three other indicators, namely, PM10, $SO_2$, and $NO_2$, which also reflect the common air quality conditions. We also included the respondents' age, gender, total years of education, number of chronic diseases, local Gross Domestic Product (GDP) per capita of the previous year, and other factors that may affect health and medical expenses as control variables.

The present study found that the air pollutants PM10, $SO_2$, and $NO_2$ affect the health status of the elderly. Among the pollutants, the impact of PM10 on health is the inverted U-shaped structure, which initially increases and then declines. Air pollution, physical health, and medical insurance costs have a significant conduction mechanism, wherein air pollution on physical health showed a highly significant positive effect; thus, more serious air pollution implies worse health status. Consequently, changes in health positively promote the increase in medical insurance costs. At the same time, to explore the difference of the effect on the elderly who have different diseases, this study also conducted an in-depth analysis on the impact of air pollution on health by dividing people into groups of those who have hypertension, heart disease, and other chronic diseases. The results show that PM10 and $SO_2$ have a significant effect on the health of patients with hypertension, whereas $SO_2$ has a greater impact on the health of those who have cardiovascular diseases. In terms of air pollution affecting the health of the population, further tests indicated that the main cause shows a distinct difference in north and south of China; hence, the pollutants $NO_2$ and $SO_2$ are harmful to the health of those in the south and north, respectively.

Based on the 15,892 sample units in China, we conducted a highly objective and quantitative assessment of the damage of air pollution on human health, which could help in understanding its true degree and provide a scientific basis, as well as serve as guidance for formulating suitable policies on balancing economic development and environmental protection in developing countries.

## 2. Theoretical Analysis and Research Hypotheses

In the past decade, air pollution caused by poor weather conditions, such as smog, gale, and sandstorm, has severely threatened the health of the middle-aged and old Chinese people increasingly. According to the public health data from the Ministry of Health (2016), respiratory diseases in middle-aged and old Chinese people have increased significantly over the past 10 years (2016 Ministry of Health data.). Health insurance costs reflect the seniors' awareness of physical and mental health insurance. In fact, an inherent mechanism of association and linkage is found among air pollution, people's health condition, and medical expenses.

Air is one of the environmental elements on which mankind depends, and it is directly involved in human life activities, including metabolism and thermoregulation. Once the severity of air pollution continues to increase, its impact on the human body will appear from recessive to dominant, which is a long accumulation process. For example, inhaled particles will accumulate and cause respiratory and pulmonary diseases through the lungs and trachea, which can also drastically stimulate eye conjunctiva, nasal cavity, and respiratory mucosa, causing lung inflammation or edema. $NO_2$ may also lead to severe pulmonary edema. These indications about air pollution are closely related to the

increasing incidence of respiratory and circulatory diseases and mortality. Therefore, the cumulative effect of air pollution on human health will have a significant impact. Hence, we hypothesized the following:

**Hypothesis 1.** *A close positive correlation exists between air pollution and individual health in the area, that is, the more serious air pollution is, the worse the health condition will be.*

Generally, health status can directly affect personal medical expenses. That is, the worse the health condition is, the greater the prevalence will be, which will then cause greater expenditure on body care, medicine, and medical insurance, and vice versa. Furthermore, air pollution can be regarded as a source of retrospective health and hygiene problems, because of the special effects on human health and its feature of long-term and large-scale effects. We speculated that the deterioration of health caused by air pollution would also lead to the change in medical insurance costs. Based on this situation, we expected the following hypothesis:

**Hypothesis 2.** *Air pollution is related to the cost of individual medical insurance costs, and the influencing path has a certain relationship with different diseases.*

## 3. Variables and Data

### 3.1. Measurement of Air Pollution and Health

AQI is used as an indicator of air pollution in most of the current research. However, the AQI index in China was officially released in 2013. Under this condition, we used the concentrations of PM10, $SO_2$, and $NO_2$ to measure the degree of air pollution. The self-rated health indicators in CHARLS database can reflect the health status of the respondents, which is the result of evaluating the health status through the angle of the individual. The indicators are divided into the following five categories: Excellent, better, good, fair, and bad. This study reprocessed these indices as follows. If the respondent was "excellent," "better," or "good," then the respondent was set to 1, whereas "fair" or "bad" were set to 0. To avoid the possible bias of using a single indicator to measure health level, we also used the "outpatient times" in four weeks and "inpatient times" within a year in the CHARLS database for comparison.

### 3.2. Selection of Control Variables

In reality, several factors affect the health status of the elderly. Hence, we selected some characteristic variables of the respondents in the CHARLS database as control variables, including age, gender, education level, and quantity of chronic disease. Moreover, the proportion of medical expenses to GDP usually increases with the development of economy and citizens' living standards. However, the level of economic development often has a certain lag effect on the demand of health services [19]. Therefore, we also selected local per capita GDP of the previous year as a control variable. The related variables are defined in Table 1.

**Table 1.** Variables and definition.

| Variable Symbol | Variable Name | Definition |
| --- | --- | --- |
| Health | Self-rated health | Excellent, Better, Good set to 1, Fair, Bad set to 0 |
| Outpatient | Outpatient frequency | Outpatient times in the past four weeks |
| Inpatient | Hospitalization times | Inpatient times in the last year |
| PM10 | Concentration of PM10 | Unit $mg/m^3$ |
| $SO_2$ | Concentration of $SO_2$ | Unit $mg/m^3$ |
| $NO_2$ | Concentration of $NO_2$ | Unit $mg/m^3$ |
| Age | Age | - |
| Gender | Gender | Female is 1, male is 0 |
| Education | Educated Years | - |
| Disease | The number of chronic diseases | - |
| Ln(GDP) | Per capita GDP | The logarithm of GDP per capita in previous year of the city |
| Medical | Medical insurance costs | Medical insurance cost last year |

### 3.3. Analysis of Descriptive Statistics

In this study, a total of 15,892 individual samples were obtained after excluding certain cities with missing relevant data of air pollution and other indicators. We first made a basic analysis of the statistical characteristics of the main variables above. The results are shown in Table 2.

**Table 2.** Descriptive statistics of main variables.

| Variable | Unit | Observations | Mean | Standard Deviation | Maximum | Minimum |
| --- | --- | --- | --- | --- | --- | --- |
| PM10 | $mg/m^3$ | 15,892 | 0.0916 | 0.0374 | 0.2450 | 0.0240 |
| $SO_2$ | $mg/m^3$ | 15,892 | 0.0313 | 0.3549 | 0.0950 | 0.0080 |
| $NO_2$ | $mg/m^3$ | 15,892 | 0.0359 | 0.0123 | 0.0700 | 0.0120 |
| Health | Dummy Variable | 15,892 | 0.6425 | 0.4792 | 1 | 0 |
| Outpatient | Frequency | 15,892 | 0.5288 | 1.6306 | 36 | 0 |
| Inpatient | Frequency | 15,892 | 0.2312 | 0.6943 | 18 | 0 |
| Age | Year | 15,892 | 61.2500 | 10.2175 | 105 | 50 |
| Education | Year | 15,892 | 5.0200 | 5.2700 | 42 | 0 |
| Diseases | Number | 15,892 | 1.0097 | 1.3205 | 10 | 0 |
| Ln (GDP) | U.S. Dollars | 15,892 | 8.8205 | 0.6334 | 10.1686 | 6.7498 |
| Gender | Dummy variable | 15,892 | 0.5300 | 0.4910 | 1 | 0 |
| Medical | CNY | 15,892 | 983.1716 | 2486.746 | 0 | 19,000 |

The descriptive analysis in Table 2 indicates that the average concentrations of the three main pollutants (i.e., PM10, $SO_2$, and $NO_2$) in the atmosphere are between 0.03–0.10 $mg/m^3$, of which the average concentration of PM10 is the highest, whereas those of $SO_2$ and $NO_2$ are similar. For the self-rated health of the dummy variable, the mean value is 0.6425, which exceeds 0.5; thus, the physical health status of the study sample is slightly better than the general level. From the statistical results of the two variables containing inpatient and outpatient treatments, the means are 0.2312 and 0.5288, respectively; hence, the number of outpatient treatment is significantly higher than that of inpatient treatment, which accords with the actual situation based on experience.

In addition, for a total of 15,892 samples over a three-year period, the average age is 61.25 years, of which 7390 (46.5%) are male and 8502 (53.5%) are female. The distribution of education years ranges from 0 to 42, with a mean of 5.02 and a standard deviation of 5.27; these values indicate that the level of education significantly differs in this study. GDP per capita in US dollars can objectively reflect the level of social economy development and the degree of development. The average medical insurance cost is 983.1716 yuan, ranging from 0 to 19,000, which has a large standard deviation.

Wooldridge put forward that, there are many advantages of using logarithms of strictly positive variables: (1) Interpretation of coefficients is easier: independent of the units of measurements of xs (elasticity or semi-elasticity); (2) when y > 0, log(y) often satisfies CLM assumptions more closely than y in levels. Strictly positive variables (prices, income, etc.) often have heteroscedastic or skewed distributions. Taking logs can mitigate these problems; (3) log transformation reduces or eliminates

skewness and reduces variance; (4) taking logs narrows the range of the variable leading to less sensitive estimates to outliers (extreme observations) [20]; for better data visualization and easier statistical inference, we took the logarithmic treatment of the variable GDP per capita and quantified the level of education.

## 4. Empirical Results and Analysis

### 4.1. Air Pollution and Self-Rated Health

We built a logistic regression model (1) in order to analyze the relationship between air pollution and self-rated health.

The logistic regression model was established as follows:

$$p_i = \frac{1}{1 + e^{-z_i}}$$

And

$$z_i = \beta_0 + \sum_{j=1}^{m} \beta_j F_{ij} + \varepsilon_i \tag{1}$$

in which

$F_{ij}$ are the factors influencing self-rated health, $\beta_j$ is the coefficient to be determined. The function of $p$ is s-type distributed and is an increasing function. $p \in (0, 1)$.

$$\lim_{z \to \infty} p = \lim_{z \to \infty} 1 + \frac{1}{e^{-z}} = 1$$

$$\lim_{z \to -\infty} p = \lim_{z \to -\infty} 1 + \frac{1}{e^{-z}} = 0$$

For each sample $i$, ($i = 1, 2, \ldots, n$), if $p \approx 0$, it indicates that the health of the body is poor, if $p \approx 1$, it indicates the health condition is good.

Let

$$p_i(y_i) = p_i^{y^i}(1 - p_i)^{(1 - y_i)}$$

in which

$$y_i = \begin{cases} 1, health\,of\,the\,the\,sample\,i\,is\,poor \\ 0, health\,of\,the\,sample\,is\,good \end{cases}$$

We took the maximum likelihood function method to find the parameters, n individuals are independent, then the joint density likelihood function of the sample is:

$$L = \prod_{i=1}^{n} p_i = \prod_{i=1}^{n} p_i^{y_i}(1 - p_i)^{(1 - y_i)}$$

Take the logarithm on both sides:

$$\begin{aligned} lnL &= ln \prod_{i=1}^{n} p_i^{y_i}(1 - p_i)^{(1 - y_i)} \\ &= \sum_{i=1}^{n} [y_i ln \frac{p_i}{1 - p_i} + ln(1 - p_i)] \\ &= \sum_{i=1}^{n} y_i z_i - ln(1 + e^{z_i})] \end{aligned}$$

in which

$$z_i = \beta_0 + \sum_{j=1}^{m} \beta_j F_{ij} + \varepsilon_i$$

Maximize the above function value, find the coefficient ($j = 0, 1, 2, \ldots, m$), and find the partial derivative and equal to 0:

$$\frac{\partial L}{\partial \beta_0} = \sum_{i=1}^{n} \left[ y_i - \frac{1}{1+e^{-z_i}} \right] = 0$$

$$\frac{\partial L}{\partial \beta_j} = \sum_{i=1}^{n} \left[ y_i - \frac{1}{1+e^{-z_i}} \right] * F_{ij} = 0$$

The above equations are combined to find the estimated parameter values.

In our model, in all pentameters $\beta_i$, we used $\beta_1$ and $\beta_2$ to represent the coefficients of the first term about the pollutant variable, and the quadratic term about the pollutant variable respectively. Thus,

$$\frac{dHealth}{dAP} = \beta_1 + 2\beta_2 AP$$

where the dependent variable *Health* represents self-rated health, which measures the health condition of each individual. As previously mentioned, if the indicator "self-rated health" is "excellent," "better," or "good," then we set it to 1; otherwise, it is 0. We used *AP* as air pollution variable to represent the concentrations of PM10, $SO_2$, and $NO_2$. In view of the possible nonlinear relationship between air pollution and physical health, we also added the quadratic term of AP into the model.

The impact from air pollution to self-rated health can be divided into four categories according to the sign of $\beta_1$ and $\beta_2$. (1) If $\beta_1 > 0$ and $\beta_2 > 0$, then the impact of air pollution on health is positive; (2) if $\beta_1 < 0$ and $\beta_2 < 0$, then the impact of air pollution on health is negative; (3) if $\beta_1 < 0$ and $\beta_2 > 0$, then the effect of air pollution on health is "initially negative and then positive"; (4) if $\beta_1 > 0$ and $\beta_2 < 0$, then the effect of air pollution on health will appear as "initially positive and then negative." Of course, this is only a theoretical way for air pollutants to affect human health, and it needs to be analyzed in combination with the air pollution concentration range. In addition, *x* represents a series of control variables that can control the population characteristics of a region.

The current micro-survey data in CHARLS database only include the years 2011, 2013, and 2015. Thus, we merged the three-year data to perform mixed logistics regression. The regression results are shown in Table 3.

We first analyzed the regression results of air pollution on self-rated health, focusing on the coefficients of the three main air pollutants in the models.

The first five columns in Table 3 show the results of regression with only one type of air pollutant concentration without control variables in the model. Let us consider the first column as an example, the first and square terms of PM10 pass the test at the significance level of 1%; the coefficients of the first and square terms are positive and negative respectively, indicating that the impact of air pollution on self-rated health seems to present an "inverted U-shaped" path, which implies an "initially positive and then negative" effect. That is, people's health condition become better at first, and then worsens with the increase in pollutant concentration, which reflects the path of impact caused by air pollution on health to some extent. However, the influence degree of various air pollutants indicates that the regression coefficients of $SO_2$ and $NO_2$ are relatively small, whereas the absolute value of PM10 coefficient is greater and has more impact on health. This impact may be due to the fact that China in recent years has weak restrictions on PM10 emission and inhalable particles compared with sulfides and nitrogen oxides, resulting in a large content of this pollutant in the air; moreover, PM10 is presently more harmful to health.

The fourth column in Table 3 shows the regression results of adding all the first terms about air pollutants into the model. The results show that the coefficients between health and $NO_2$ and $SO_2$ pass the test at the 1% significance level, whereas PM10 is significant at 10%. After adding the first and square terms of the three air pollutant concentrations, the regression results are shown in the fifth column of Table 3, in which only PM10 shows a statistical significance. The first and second coefficients are positive and negative respectively, which is consistent with the first regression result. $NO_2$ and $SO_2$ do not show significant effects.

**Table 3.** Effect of air pollution on self-rated health (self-rated health as the dependent variable).

| Variable | Regression (1) | Regression (2) | Regression (3) | Regression (4) | Regression (5) | Regression (6) | Regression (7) | Regression (8) | Regression (9) | Regression (10) |
|---|---|---|---|---|---|---|---|---|---|---|
| PM10 | 11.6392*** (9.3200) | | | 4.2043*** (8.0000) | 0.2644*** (9.3400) | 0.1050*** (4.3100) | | | 0.1370*** (6.1300) | 0.1990*** (6.7300) |
| PM10$^2$ | −25.3428*** (−5.6900) | | | ____ | −0.0462*** (−5.6900) | −0.0170** (−2.0900) | | | | −0.0350*** (−4.1100) |
| SO$_2$ | ____ | 0.0618** (2.4400) | | 0.0242*** (3.3500) | 0.0187 (0.7300) | | 0.1088 (1.2900) | | 0.0854*** (3.4600) | 0.0729 (0.8500) |
| SO$_2{}^2$ | ____ | −0.0006 (−1.2100) | | ____ | −7.38 × 10$^{-6}$ (−0.0100) | | −0.0022 (−0.3900) | | | −0.0006 (−0.1000) |
| NO$_2$ | ____ | ____ | 0.1474*** (8.9400) | 3.5046** (2.0500) | 0.0096 (0.4400) | | | −0.0554*** (−2.9800) | −0.1327*** (-5.8100) | −0.1546*** (−6.5700) |
| NO$_2{}^2$ | ____ | ____ | 0.0708*** (5.1700) | ____ | 0.0579*** (4.0100) | | | 0.0761*** (5.4700) | | 0.0628*** (4.2900) |
| Age | ____ | ____ | ____ | ____ | ____ | −0.0252 (−1.5200) | −0.0280* (−1.6900) | −0.0304* (−1.8400) | −0.0288* (−1.7300) | −0.0293* (−1.7600) |
| Education | ____ | ____ | ____ | ____ | ____ | 0.1512*** (8.4300) | 0.1514*** (8.4900) | 0.1574*** (8.7900) | 0.1542*** (8.5800) | 0.1584*** (8.7800) |
| Disease | ____ | ____ | ____ | ____ | ____ | −0.3439*** (−20.96) | −0.3417*** (−20.9800) | −0.3418*** (−20.8700) | −0.3442*** (−20.9500) | −0.3452*** (−20.9200) |
| Ln (GDP) | ____ | ____ | ____ | ____ | ____ | 0.3167*** (17.5700) | 0.3457*** (20.7300) | 0.3716*** (19.7800) | 0.3623*** (19.3000) | 0.3554*** (18.7200) |
| Gender | ____ | ____ | ____ | ____ | ____ | −0.0509 (−1.5000) | −0.0458 (−1.3600) | −0.0437 (−1.2900) | −0.0518 (−1.5300) | −0.0500 (−1.4700) |
| C | −0.2688 | 0.5396 | 0.4831 | 0.0302 | 0.5447 | 0.6256 | 0.6058 | 0.5263 | 0.6117 | 0.5836 |

Note: (1) We use mixed logistics regression to estimate Model (1). We use the ordinary likelihood ratio test to select a model that is better suited to current data analysis. (2) "z value" is enclosed in parentheses; *, **, and *** denote the levels of significance by 10%, 5%, and 1%, respectively. (3) We use coldaig2 to test the multicollinearity problems. Result for each regression are shown in Appendix C.1 Multicollinearity test for logistic regression. For those regressions which didn't pass the test, we conducted modification one by one.

Finally, to measure the impact of air pollution on health accurately and compare with the regression results above, we controlled the age, education years, number of chronic diseases, and regional GDP per capita. The related results are listed in the last five columns in Table 3. The results of adding the three air pollutants separately into the model indicate that PM10 and $NO_2$ pass the significance test at the levels of 1%, whereas $SO_2$ is not significant. Meanwhile, when the three pollutants are added into the model all at once, the coefficients of PM10, $NO_2$ and $SO_2$ are all significant at the level of 1%, indicating that the results are better after adding these three pollutants into the model together. Moreover, similar results are obtained in the nonlinear model with the first and second term additions of the three air pollutant concentrations. The primary and secondary terms of PM10 and $NO_2$ pass the test at the significance level of 1%, whereas $SO_2$ is not significant.

It is worth mentioning that: The coefficient of PM10 is 11.6392 in Regression (1) and 0.1050 in Regression (6) in Table 3, which is mainly due to the control of more variables, which reduces the explanatory power of explanatory variables and makes them closer to the real situation. Refer to the above for an explanation of the relevant situation later.

To sum up, Hypothesis 1 indicates that the air pollution status is negatively related to the health of individuals in the region to a certain extent. That is, the more severe air pollution is, the worse health condition will be.

### 4.2. Self-Rated Health and Medical Insurance Costs

To verify whether a significant change exists in medical expenses under different health conditions, and to make a reference and comparison with the following study about the impact of air pollution on medical insurance costs, we established an OLS regression Model (2) as

$$Medical \ = \propto +\beta_1 health + \gamma X + \varepsilon \tag{2}$$

where the dependent variable *Medical* is an indicator of individual medical insurance costs, and $X$ represents a series of control variables in terms of the population characteristics of a region. We also performed a hybrid cross-sectional regression using the three-year data in the CHARLS database. The estimated results are shown in Table 4.

**Table 4.** Effect of self-rated health on medical insurance costs.

| Variable | OLS Regression |
|----------|----------------|
| Health | −102.2384 ** (−2.4800) |
| Ln GDP | −18.3292 (−0.6000) |
| Disease | 168.6429 *** (13.3800) |
| Education | 3.8546 (1.0100) |
| Gender | −32.0054 (−0.8000) |
| Age | −0.3503 (−0.1800) |
| C | 1061.1400 |

Note: (1) "*t* value" is enclosed in parentheses; ** and *** denote the levels of significance test by 10%, 5%, and 1%, respectively; (2) we used mixed OLS regression to estimate Model (2); (3) we used White's (1980) test in the regression analysis to test possible heteroscedasticity problems. The detailed test process is shown in Appendix B, and the same as the following Tables; (4) we used VIF to test the multicollinearity problems. Results for each model are shown in Appendix C.2. Multicollinearity test for OLS regression; (5) we conducted residential analysis, and the residual plots are shown in Appendix D Residential Analysis. For those regressions which did not pass the test, we conducted modification one by one.

In view of other relevant factors that influence medical expenses, we added a series of variables that represent demographic characteristics, including age, gender, number of chronic diseases, local GDP and educational years. We selected local per capita GDP of the previous year as a control variable due to the lagged effect of economic level on the demand for health services.

From Table 4, the regression coefficient of self-rated health is −102.2384, which indicates that individuals will generally increase their medical insurance costs to obtain premium payments in the

event of a medical condition to reduce personal medical expenses, because they are aware of the deterioration of their health status. Hence, the empirical results of self-rated health status are in line with expectations, and the result is statistically significant. Among all the control variables, the chronic disease index passes the test at a significance level of 1%, which shows that the individual health status (the index of chronic disease type) is closely related to the local medical insurance costs.

### 4.3. Mechanism of Air Pollutants Affecting Medical Insurance Cost

To further clarify the operated mechanism between air pollution, self-rated health, and medical insurance costs, we added the interaction of air pollution and self-rated health as an explanatory variable and made a comparative analysis with the self-rated health in Regression Model (2). Thus, we built the OLS regression Models (3), (4) and (5) as follows:

$$Medical = \alpha + \beta_1 health + \beta_2 AP + \gamma X + \varepsilon \tag{3}$$

$$Medical = \alpha + \beta_1 health + \beta_2 AP \times health + \gamma X + \varepsilon \tag{4}$$

$$Medical = \alpha + \beta_1 health + \beta_2 AP \times health + \beta_3 AP + \gamma X + \varepsilon \tag{5}$$

where the independent variable *AP* represents the concentrations of air pollutants, PM10, $SO_2$, and $NO_2$. To verify the effect of air quality on medical insurance costs by indirectly influencing people's health, we added the interaction of air quality and health to test the indirect impact of air quality on medical insurance costs. The empirical results are shown in Table 5.

**Table 5.** Regression results of Models (3), (4) and (5) on air pollution, physical health, and medical insurance costs.

| Variable | Medical Insurance Costs | | |
|---|---|---|---|
| | **No Interaction Variables** | **No separate Air Pollutant Variables** | **Adding Individual Air Pollutant Variables** |
| PM10 | −2061.9410 *** (−3.7400) | —— | −2461.5130 * (−2.5900) |
| PM10 * health | —— | −1753.2700 *** (−2.6000) | 731.5704 (0.6200) |
| $SO_2$ | −20.9259 (−1.1000) | —— | −28.6356 (−0.5700) |
| $SO_2$ * health | —— | −20.6674 (−0.9900) | 4.1824 (0.1500) |
| $NO_2$ | −3523.4990 (−1.6000) | —— | −3195.8220 (−0.9200) |
| $NO_2$ * health | —— | −3103.6500 (−1.1900) | −715.3406 (−0.1700) |
| Health | −91.3408 ** (−2.1700) | 184.8721 ** (2.0600) | −140.0211 (−1.0600) |
| Age | −0.2830 (−0.1390) | −0.2602 (−0.1300) | −0.5559 (−0.2800) |
| Gender | −28.6215 (−0.7100) | −32.2771 (−0.8000) | −30.2722 (−0.7400) |
| Education | 4.7503 (1.2400) | 3.6234 (0.9700) | 5.9962 (1.5600) |
| Diseases | 150.4636 *** (12.2100) | 157.0992 *** (12.5800) | 150.8624 *** (12.1400) |
| Ln (GDP) | 37.2540 (1.0600) | 10.2010 (0.3000) | 38.1661 (1.0900) |
| C | 910.8777 | 828.2811 | 942.2200 |

Note: (1) "*t* value" is enclosed in parentheses; *, **, and *** denote the levels of significance test by 10%, 5%, and 1%, respectively; (2) we use mixed OLS regression to estimate Model (2); (3) we used White's (1980) test in the regression analysis to test possible heteroscedasticity problems. The detailed test process is shown in Appendix B, and the same as the following Tables; (4) we used VIF to test the multicollinearity problems. Results for each model are shown in Appendix C.2. Multicollinearity test for OLS regression; (5) we conducted residential analysis, and the residual plots are shown in Appendix D Residential Analysis. For those regressions which did not pass the test, we conducted modification one by one.

The results show that when air pollution is added into the model and the interaction term between air pollution and self-rated health is not added, the concentration of air pollution and self-rated health are inversely related to the expenditure of medical insurance. When air pollution, the interaction between air pollution and self-rated health are both added into the model, the self-rated health has a positive impact on the expenditure of medical insurance, but air pollution still has a negative impact on the expenditure of medical insurance on the whole. It seems that self-rated health is inversely proportional to medical insurance expense, but after removing the influence of air

pollutant concentration on medical insurance expense, self-rated health on medical insurance expense is positively correlated, that is, the higher the self-rated health level, the higher the medical insurance expense should be, which is reasonable, because medical insurance requires that the insured person is in good health. This proves that air pollution can affect medical insurance expense by affecting self-rated health, which is consistent with Hypothesis 2.

Only the number of chronic diseases negatively affects medical insurance expense at the significance level of 1%, and the parameters of other variables are not very significant. The reason for this result may lie in the fact that; (1) the major factors currently affecting the medical insurance expenses in China are mainly diseases, especially those with long treatment cycles. In this regard, the data released by the Chinese Ministry of Health also seems to support this point; (2) moreover, the effect of air pollution on self-rated health and medical insurance expenses of micro-entities may have a certain degree of lag; thus, the impact on medical insurance expenses in the current period will not be significant; (3) furthermore, the short-term effect is not evident due to the limited time available to obtain individual data.

*4.4. Further Study*

Air pollution affects the health of middle-aged and old people. However, is there any difference in the impact among people who have different diseases? Local scholars have shown that air pollution leads to respiratory and cardiovascular diseases, which increases the rate of system incidence, visiting, admission, and mortality [21–24]. Short-term air pollution exposure increases the risk of respiratory and cardiovascular diseases in healthy people [25–27].

Heart disease and hypertension (high blood pressure) are among the three most common chronic diseases in the elderly (see Appendix A). We selected the similar prevalence of heart disease and hypertension as a sub-sample basis to explore the different populations affected by air pollution (Red as above).

A straightforward assumption is that the impact caused by air pollution on the health conditions of people with different diseases is different. We constructed OLS regression Models (6) and (7) to verify the effect of air pollution on health in different affected populations.

$$Health = \alpha + \beta_1 age + \beta_2 education + \beta_3 disease + \beta_4 ln(gdp) + \beta_5 gender \quad (6)$$

$$Health = \alpha + \beta_1 PM10 + \beta_2 PM10^2 + \beta_3 SO_2 + \beta_4 SO_2^2 + \beta_5 NO_2 + \beta_6 NO_2^2 \quad (7)$$

Table 6 shows that the rate of hypertension, age, education years, number of chronic diseases, and local GDP in the previous year all pass the test with a significance of 1%. Among the factors, self-rated health and age of the patients are statistically significant ($p < 0.01$), and the regression coefficient is $-0.1299$. Therefore, as age increases, the health condition of patients with hypertension continues to deteriorate. In addition, the health of patients with hypertension improves with the development of local GDP level, reflecting that the local economic level and medical services can effectively reflect the health of patients. The patients' health and education level also pass the test at a significance level of 1% with a regression coefficient of 0.1677, indicating that a higher education level indicates better health condition. Meanwhile, the results of patients with heart diseases show a certain similarity: Self-Rated health and the number of chronic diseases, the local GDP level as well as gender are statistically significant ($p < 0.01$). It is indicated that female patients with heart disease have better self-rated health than male patients. Statistical significance among the patients' age, education level, and health, which may be related to the influence from genetic factors to the incidence of heart disease, is not observed.

**Table 6.** Regression result between basic information and self-rated health in patients with hypertension and heart diseases.

| Variable | Self-Rated Health | |
| --- | --- | --- |
| | **Hypertension** | **Heart Disease** |
| Age | −0.1299 *** (−2.7300) | −0.0206 (−0.2900) |
| Education | 0.1677 *** (3.3400) | 0.0703 (0.9400) |
| Diseases | −0.1494 *** (−3.1900) | −0.2565 *** (−3.5700) |
| Lngdp | 0.3879 *** (8.0200) | 0.3373 *** (4.5900) |
| Gender | −0.0119 (−0.1200) | 0.3395 ** (2.3400) |
| Adj.R2 | 0.2095 | 0.0208 |

Note: (1) We used mixed logistics regression to estimate Model (1). We used the ordinary likelihood ratio test to select a model that is better suited to current data analysis; (2) "z value" is enclosed in parentheses; **, and *** denote the levels of significance by 10%, 5%, and 1%, respectively; (3) we used coldaig2 to test the multicollinearity problems. Results for each regression are shown in Appendix C.1. Multicollinearity test for logistic regression. For those regressions which did not pass the test, we conducted modification one by one.

As shown in Table 7 and Figure 1, the self-rated health of patients with hypertension and PM10 show a statistical significance ($p < 0.01$); moreover, linear regression coefficient of PM10 is 0.4164, whereas the square term regression coefficient is −0.0368, indicating that the impact of PM10 on the health of patients with hypertension shows an "initially positive and then negative" effect, whereas the opposite result occurs in $SO_2$, that is, "initially negative and then positive" effect. Among the coefficients, the first order of $SO_2$ is 77.5688 and the coefficient of quadratic term is 1491.7310, both of which have statistical significance, that is, the positive U−shaped effect of "initially negative and then positive," and the regression results with the overall sample show some differences. However, no significant effect between $NO_2$ and the health of patients with hypertension is observed.

**Table 7.** Regression result between air pollution and self-rated health in patients with hypertension and heart diseases.

| Variable | Hypertension | Heart Disease |
| --- | --- | --- |
| PM10 | 0.4164 *** (5.7600) | 0.3616 *** (3.0000) |
| PM10$^2$ | −0.0368 (−1.3600) | −0.0607 (−1.4400) |
| SO$_2$ | 77.5688 * (1.7400) | 10.2429 (0.6200) |
| SO$_2$$^2$ | 1491.7310 ** (2.4600) | 412.9760 (1.6300) |
| NO$_2$ | −0.0059 (−0.1100) | −0.0281 (0.3200) |
| NO$_2$$^2$ | 0.0018 (0.0400) | 0.0097 (0.1500) |
| Adj.R2 | 0.9210 | 0.0115 |

Note: (1) We used mixed logistics regression to estimate Model (1). We used the ordinary likelihood ratio test to select a model that is better suited to current data analysis; (2) "z value" is enclosed in parentheses; *, **, and *** denote the levels of significance by 10%, 5%, and 1%, respectively; (3) we used coldaig2 to test the multicollinearity problems. Result for each regression are shown in Appendix C.1. Multicollinearity test for logistic regression. For those regressions which did not pass the test, we conducted modification one by one.

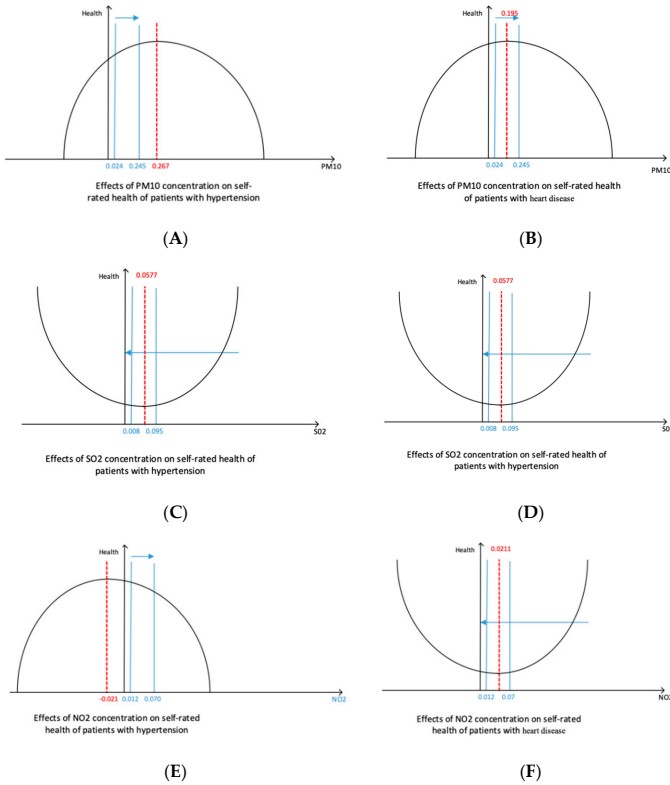

**Figure 1.** Effects of $NO_2$, PM10 and $SO_2$ concentration on self-rated health of hypertension and heart disease.

The impact of air pollutants on the health of patients with heart diseases is not significantly different from those with hypertension. Among the impacts, the square and first terms of $SO_2$ not pass the regression under the significance level of 10%, the quadratic coefficient is 412.9760, and the coefficient of first term is 10.2429. Therefore, the effect of $SO_2$ on self-rated health may be "the positive U−shaped." The coefficient of PM10 is 0.3616, which is significant. A significant effect of the $NO_2$ coefficient is not observed.

The inflection point of air pollution on self-rated health is shown in Table 8. The collected $NO_2$ concentration range has no inflection point (not shown in the table above) because of the limitations of the sample selected in this study, that is, the effect on health becomes severe as its concentration increases.

**Table 8.** Inflection point of the impact from air pollutants to Self-Rated health of patients with hypertension and heart diseases (Unit: mg/$m^3$).

| Pollutants | Concentration Range | Hypertension | Heart Disease |
|---|---|---|---|
| PM10 | 0.0240–0.2450 | 0.2670 | 0.1950 |
| $SO_2$ | 0.0080–0.0950 | 0.0577 | 0.0571 |
| $NO_2$ | 0.0120–0.0700 | −0.0213 | 0.0211 |

A meta−analysis of heart failure and air pollution shows a positive correlation between $SO_2$ concentration, and hospitalization and mortality in heart failure [24]. Combined with the regression results of air pollutants and self-rated health of patients with hypertension and heart diseases, $SO_2$ has an inflection point on the health of patients with heart diseases. That is, as the $SO_2$ concentration increases, the impact of health is initially negative, but becomes positive when it reaches 0.014 mg/$m^3$. For PM10, the effect on the health of patients with hypertension becomes negative when the level of

0.040 mg/m$^3$ is reached. When the level of 0.064 mg/m$^3$ is reached, the effect on the health of patients with heart diseases is negative and can be regarded as the harmful concentration of PM10.

## 5. Robustness Test

### 5.1. Adding Outpatient Frequency and Hospitalization Times

To ensure the robustness of the empirical results, we used outpatient frequency and hospitalization times, which can reflect individual health indicators to replace the self-rated health as dependent variables in the regression model. The results are shown in Tables 9 and 10.

The regression results using outpatient frequency as dependent variable are shown in Table 9. The first three columns show the results with no control variables. PM10 and NO$_2$ are significant, and the sign is consistent with the expectations, whereas SO$_2$ is not significant. Meanwhile, regressions from 6 to 8 are the empirical results after adding the control variables on the basis of the regressions from 1 to 3. The regression results are almost identical to the first three regressions, that is, PM10 and NO$_2$ pass the significance test. One possible factor may be that PM10 and NO$_2$ are more harmful to health than the other one pollutant during this particular period.

Column 4 in Table 9 shows the result of regression, including all the first terms of PM10, SO$_2$, and NO$_2$ as independent variables, in which SO$_2$ is more significant. Similarly, the result of regression 9 after adding all the control variables based on Regression Model 4 is also consistent, which indicates that SO$_2$ also has a significant positive effect on individual health; however, the effect is still less than that of PM10 and NO$_2$, considering their value of regression coefficient.

Moreover, we regress the first and square terms of PM10, SO$_2$, and NO$_2$ as independent variables and consider any significant change of the results without control variables. The empirical results are shown in regressions 5 and 10 of Table 9. The results show that PM10 and SO$_2$ still have significant positive effects on individual health, although their effects are diminishing according to the regression coefficients.

Finally, in most regression models, the results of PM10 show an "inverted U-shaped" phenomenon; thus, the coefficients of the first and square items are positive and negative, respectively, which is the same in Table 3.

The air pollutants represented by PM10 and SO$_2$ affect the health conditions of individuals, and PM10 has the greatest impact on health, which may be because the PM2.5 index has not yet been measured separately before 2013 in China. As a result, the impact of PM10 on the respiratory system of an individual includes the effects of PM2.5, which manifests as the obvious effects of PM10. As we separately add PM2.5 indicators in a later time, the impact of PM10 on health significantly decreases. The effect of SO$_2$ on health should be related to its strong toxicity.

Table 10 shows the regression results using hospitalization times as an independent variable. Generally, the conclusion that air pollution significantly affects the health conditions of an individual remains valid. Particularly, most of the results in Table 10 are almost identical to those in Table 9; however, the significance of indicators (i.e., PM10, SO$_2$, and NO$_2$), which measure air pollution concentrations, are significantly high, and the effect is strong (with generally large regression coefficients). The NO$_2$ indicator, which is not significant in Table 9, also shows a significant positive effect on health. In addition, similar to Table 3, the effect of SO$_2$ on health is mostly manifested as "inverted U-shaped."

**Table 9.** Effect of air pollution on self-rated health (hospitalization times as independent variable).

| Variable | Regression (1) | Regression (2) | Regression (3) | Regression (4) | Regression (5) | Regression (6) | Regression (7) | Regression (8) | Regression (9) | Regression (10) |
|---|---|---|---|---|---|---|---|---|---|---|
| PM10 | 0.0153** (2.2500) | —— | —— | 0.1594 (1.0500) | 0.0127 (1.4300) | 0.0198*** (2.7100) | —— | —— | 0.2597* (1.7200) | 0.0455*** (7.1000) |
| PM10$^2$ | −0.0029 (−1.2200) | —— | —— | —— | −0.0026 (−1.0400) | −0.0019 (−0.8000) | —— | —— | —— | −0.0081*** (−4.4300) |
| SO$_2$ | —— | −0.0020 (−0.1000) | —— | 0.0039*** (2.5900) | −0.0076 (−0.3800) | —— | 0.0052 (0.2700) | —— | 0.0042*** (2.8500) | 0.0127 (0.8900) |
| SO$_2$$^2$ | —— | 0.0011 (0.8600) | —— | —— | 0.0014 (1.0500) | —— | 0.0008 (0.0601) | —— | —— | −0.0002 (−0.2500) |
| NO$_2$ | —— | —— | 0.0089* (1.7700) | 0.2202 (0.42) | 0.0007 (0.1000) | —— | —— | 0.0150*** (2.6300) | 0.5577 (0.9900) | −0.0360*** (−7.0600) |
| NO$_2$$^2$ | —— | —— | 0.0038 (0.9500) | —— | 0.0021 (0.4900) | —— | —— | 0.0056 (1.3800) | —— | 0.0131*** (4.2700) |
| Age | —— | —— | —— | —— | —— | 0.0240*** (4.7800) | 0.0239*** (4.8000) | 0.0243*** (4.8800) | 0.0024*** (4.8200) | −0.0066* (−1.8300) |
| Education | —— | —— | —— | —— | —— | −0.0198*** (−3.8400) | −0.0188*** (−3.6700) | −0.0189*** (−3.6900) | −0.0037*** (−3.7800) | 0.0326*** (8.7800) |
| Disease | —— | —— | —— | —— | —— | 0.0542*** (10.8700) | 0.0535*** (10.8300) | 0.0545*** (10.9900) | 0.0419*** (11.0500) | −0.0785*** (−21.8300) |
| Ln(GDP) | —— | —— | —— | —— | —— | −0.0027 (−0.4900) | 0.0029 (0.5800) | −0.0035 (−0.6200) | −0.0061 (−0.7000) | 0.0803*** (19.6000) |
| Gender | —— | —— | —— | —— | —— | −0.0047 (−0.4600) | −0.0042 (−0.4100) | −0.0041 (−0.4000) | −0.0045 (−0.4400) | −0.0115 (−1.5700) |
| C | 0.1886 | 0.1836 | 0.1810 | 0.1616 | 0.1849 | 0.1899 | 0.1859 | 0.1811 | 0.0278 | 0.6358 |

Note: (1) "t value" is enclosed in parentheses; *, **, and *** denote the levels of significance test by 10%, 5%, and 1%, respectively. (2) We use mixed OLS regression to estimate Model (2). (3)We use White's (1980) test in the regression analysis to test possible heteroscedasticity problems. The detailed test process is shown in Appendix B, and the same as the following Tables. (4) We use VIF to test the multicollinearity problems. Result for each model is shown in Appendix C.2 Multicollinearity test for OLS regression. (5) We conducted Residential analysis and the residual plots are shown in Appendix D Residential Analysis. For those regressions which didn't pass the test, we conducted modification one by one.

**Table 10.** Effect of air pollution on self-rated health (Outpatient frequency as independent variable).

| Variable | Regression (1) | Regression (2) | Regression (3) | Regression (4) | Regression (5) | Regression (6) | Regression (7) | Regression (8) | Regression (9) | Regression (10) |
|---|---|---|---|---|---|---|---|---|---|---|
| PM10 | −0.0157 (−1.1700) | —— | —— | −1.0662*** (−3.5700) | −0.0994*** (−5.6800) | 0.0154 (1.0600) | —— | —— | −1.0631*** (−3.5700) | −0.0812*** (−4.6100) |
| PM10$^2$ | 0.0095** (2.0507) | —— | —— | | 0.0210*** (4.2300) | 0.0015 (0.3100) | —— | | | 0.0143*** (2.8500) |
| SO$_2$ | —— | 0.0713* (1.8600) | —— | 0.0071** (2.4300) | 0.0474 (1.2100) | —— | 0.0758** (1.9700) | —— | 0.0059** (2.0100) | 0.0392 (1.0000) |
| SO$_2^2$ | —— | −0.0027 (−1.0600) | —— | | −0.0017 (−0.6400) | —— | −0.0030 (−1.1400) | —— | | −0.0013 (−0.5100) |
| NO$_2$ | —— | —— | 0.0483*** (4.8300) | 6.0536*** (5.8500) | 0.0910*** (6.8400) | —— | —— | 0.0940*** (8.3500) | 9.6786*** (8.7200) | 0.1271*** (9.0800) |
| NO$_2^2$ | —— | —— | 0.0115 (1.4500) | | 0.0076 (0.9000) | —— | —— | 0.0046 (0.5800) | | 0.0031 (0.3700) |
| Age | —— | —— | —— | —— | —— | 0.0211** (2.1300) | 0.0214** (2.1700) | 0.0241** (2.4400) | 0.0024** (2.4400) | 0.0243** (2.4500) |
| Education | —— | —— | —— | —— | —— | 0.0014 (0.1300) | 0.0024 (0.2300) | 0.0004 (0.0400) | 0.0003 (0.1500) | 0.0021 (0.2100) |
| Disease | —— | —— | —— | —— | —— | −0.0371*** (−3.7600) | −0.0368*** (−3.7600) | −0.0308*** (−3.1400) | −0.0249*** (−3.3300) | −0.0308*** (−3.1300) |
| Ln(GDP) | —— | —— | —— | —— | —— | −0.0633*** (−5.9100) | −0.0597*** (−6.0300) | −0.1009*** (−9.0900) | −0.1535*** (−9.0000) | −0.0951*** (−8.4600) |
| Gender | —— | —— | —— | —— | —— | 0.0494** (2.4400) | 0.0489** (2.4400) | 0.0485** (2.4200) | 0.0507** (2.5200) | 0.0512** (2.5400) |
| C | 0.3284 | 0.3394 | 0.3252 | 0.2165 | 0.3104 | 0.3109 | 0.3141 | 0.3070 | 1.2779 | 0.2949 |

Note: (1) "t value" is enclosed in parentheses; *, ** and *** denote the levels of significance test by 10%, 5%, and 1%, respectively. (2) We use mixed OLS regression to estimate Model (2). (3) We use White's (1980) test in the regression analysis to test possible heteroscedasticity problems. The detailed test process is shown in Appendix B, and the same as the following Tables. (4) We use VIF to test the multicollinearity problems. Result for each model is shown in Appendix C.2 Multicollinearity test for OLS regression. (5) We conducted Residential analysis and the residual plots are shown in Appendix D Residential Analysis. For those regressions which didn't pass the test, we conducted modification one by one.

No significant difference is found from the empirical results in Tables 3, 9 and 10, and the three air pollutants pass the significance test to different degrees. Hence, the impact of air pollution on health shows a significant positive effect. $SO_2$ and other indicators of air pollution influence physical health in an inverted U-shaped path as "initially positive and then negative." As shown in Tables 3–10, the sign of the quadratic coefficient of PM10 is from negative to positive, which has an "unstable" effect on self-rated health.

*5.2. Difference between Northern and Southern China*

Given the large geographical differences in China, we divided the overall sample into two, including the north and south of China according to the Qinling-Huaihe River (geographical boundary between south and north) to verify whether the empirical results will change as the sample changes. Several studies have shown that the south and north are relatively different in terms of natural conditions, agricultural production methods, geographical features, and local customs.

Table 11 shows the regression results of the samples containing 31 southern cities in China. Overall, the regression results of the southern samples are generally better than the overall sample, indicating that the regression effect has improved after excluding regional differences. Particularly, the first three columns show results without control variables, reflecting that $NO_2$ and PM10 pass the test at a significance level of 10%. $NO_2$ also exhibits a mechanism of U-shaped effect on self-rated health, whereas the PM10 coefficient shows the opposite influence compared with $NO_2$. After adding the first term of three pollutants simultaneously, except for $SO_2$, the other two pollutants pass the test at a significance level of 1%, indicating a difference from the result of the overall sample due to the regression of $SO_2$. We initially deduced that the percentages of the three air pollutants in the north and south are relatively different, resulting in differences in the regression results between the overall and southern samples. This difference is reflected in $SO_2$, whereas PM10 is significant in the overall and southern regression results.

The last five columns of Table 11 show the regression results after adding the control variables. Among the regressions to which the three pollutants are added, only the first and square terms of $NO_2$ and PM10 pass the test at a significance level of 1%, whereas $SO_2$ is not significant. The regression results of PM10 and $NO_2$ are consistent with the regression results of the overall sample. Moreover, PM10, $NO_2$, and $SO_2$ pass the significance test when the first term of the three pollutants are added simultaneously, which is in accordance with the characteristics of the overall sample.

The difference between the regression results of 25 cities in northern China and 31 cities in the south are shown in Table 12. Overall, among the 10 regressions listed, the significance of the air pollutants, PM10, $SO_2$, and $NO_2$, on self-rated health indicators is enhanced, exceeding the regression results of the overall and southern samples. Specifically, in the separate regression of the three pollutants without control variables, only the first term of $NO_2$ does not show any significance, whereas the other indicators pass the regression at the significance level of 10%. After controlling other influencing variables, PM10 and $SO_2$ show a high significance (1%) on self-rated health, whereas $NO_2$ is less significant.

Thus, no difference is found between the regression results of samples from the north and south and the overall sample. However, between the north and south samples, the types of major pollutants that are hazardous to health show a difference. Among the northern samples, the significance of air pollutants on self-rated health indicators is better than that of the southern and overall samples; hence, the north has poor air quality due to burning coal for heating and other influence, resulting to the relatively poor health status of local residents. That is, the impact of air pollution on physical health is more significant. In terms of pollutant types, PM10 accounts for a larger proportion in the north and south, causing inevitable adverse effects on health conditions. $NO_2$ mainly exists in the air pollutants in the south, which should be their focus of air pollution control. $SO_2$ is one of the pollutants that is harmful to people's health in the north, which may also point the direction for key targets in effective air pollution control in northern and southern China in the future.

**Table 11.** Effect of air pollution on self-rated health in 31 southern cities of China.

| Variable | Regression (1) | Regression (2) | Regression (3) | Regression (4) | Regression (5) | Regression (6) | Regression (7) | Regression (8) | Regression (9) | Regression (10) |
|---|---|---|---|---|---|---|---|---|---|---|
| PM10 | 17.8429*** (2.0764) | ____ | ____ | 4.7140*** (0.8734) | 0.2609*** (7.8800) | 0.1115*** (3.4800) | ____ | ____ | 0.1237*** (4.3900) | 0.1560*** (4.4600) |
| PM10$^2$ | −52.8684*** (8.7510) | ____ | ____ | ____ | −0.0846*** (−7.7300) | −0.0207** (−2.0400) | ____ | ____ | ____ | −0.0370*** (−3.1000) |
| SO$_2$ | ____ | 0.0570** (0.0252) | ____ | 0.0199*** (0.0072) | 0.0359 (1.2900) | ____ | 0.1322 (1.2000) | ____ | 0.1005*** (3.1100) | 0.0639 (0.5600) |
| SO$_2{}^2$ | ____ | −0.0005 (0.0005) | ____ | ____ | 0.1105*** (4.9300) | ____ | −0.0034 (−0.3400) | ____ | ____ | 0.0006 (0.0600) |
| NO$_2$ | ____ | ____ | −16.2336* (8.2883) | 4.3751** (2.1504) | 0.0053 (0.0500) | ____ | ____ | −0.0921*** (−3.3800) | −0.1385*** (−4.6000) | −0.1449*** (−4.7000) |
| NO$_2{}^2$ | ____ | ____ | 394.2988*** (109.935) | ____ | 0.0040 (0.4000) | ____ | ____ | 0.0818*** (4.6100) | ____ | 0.0835*** (3.7100) |
| Age | ____ | ____ | ____ | ____ | ____ | −0.0316 (−1.4400) | −0.0325 (−1.4900) | −0.0347 (−1.5900) | −0.0330 (−15000) | −0.0345 (−1.5700) |
| Education | ____ | ____ | ____ | ____ | ____ | 0.1554*** (6.4500) | 0.1504*** (6.3100) | 0.1567*** (6.5400) | 0.1610*** (6.6500) | 0.1649*** (6.8000) |
| Disease | ____ | ____ | ____ | ____ | ____ | −0.3217*** (−15.0200) | −0.3152*** (−14.8900) | −0.3161*** (−14.8300) | −0.3212*** (−14.9800) | −0.3212*** (−14.9500) |
| Ln(GDP) | ____ | ____ | ____ | ____ | ____ | 0.3059*** (12.6800) | 0.3395*** (15.5000) | 0.3745*** (15.1700) | 0.3615*** (14.6900) | 0.3405*** (12.9500) |
| | ____ | ____ | ____ | ____ | ____ | −0.0464 (−1.0300) | −0.0416 (−0.9300) | −0.0396 (−0.8800) | −0.0460 (−1.0200) | −0.0446 (−0.9900) |
| C | −0.4249 | 0.5858 | 0.6397 | 0.0772 | 0.5878 | 0.6834 | 0.6585 | 0.5713 | 0.6646 | 0.6173 |

Note: (1) We use mixed logistics regression to estimate Model (1). We use the ordinary likelihood ratio test to select a model that is better suited to current data analysis. (2) "z value" is enclosed in parentheses; *, **, and *** denote the levels of significance by 10%, 5%, and 1%, respectively. (3) We use coldaig2 to test the multicollinearity problems. Result for each regression are shown in Appendix C.1 Multicollinearity test for logistic regression. For those regressions which didn't pass the test, we conducted modification one by one.

**Table 12.** Effect of air pollution on self-rated health in 25 northern cities of China.

| Variable | Regression (1) | Regression (2) | Regression (3) | Regression (4) | Regression (5) | Regression (6) | Regression (7) | Regression (8) | Regression (9) | Regression (10) |
|---|---|---|---|---|---|---|---|---|---|---|
| PM10 | 21.1412*** (2.1062) | —— | —— | 4.8368*** (0.9669) | 0.4641*** (7.6000) | 0.3389*** (7.8000) | —— | —— | 0.2283*** (4.6800) | 0.5010*** (7.8600) |
| PM10$^2$ | −47.7789*** (6.4352) | —— | —— | —— | −0.2171*** (−7.5300) | −0.1002*** (−6.2200) | —— | —— | —— | −0.2128*** (−7.2600) |
| SO$_2$ | —— | 11.3783** (5.2098) | —— | −1.4041 (2.1293) | −27.3279*** (−3.6200) | —— | 0.0581 (1.4600) | —— | −0.1112** (−2.4200) | −0.0129 (−0.2500) |
| SO$_2{}^2$ | —— | −1.3500 (47.0390) | —— | —— | 362.0244*** (4.3800) | —— | 0.0011 (0.0600) | —— | —— | 0.1130*** (3.3200) |
| NO$_2$ | —— | —— | −0.5982 (15.2933) | 13.4551*** (3.3509) | −0.0501 (−1.0100) | —— | —— | 0.0998*** (2.9200) | 0.0273 (0.6600) | −0.2061*** (−3.7900) |
| NO$_2{}^2$ | —— | —— | 336.6407* (198.8603) | —— | −0.0342 (−1.3100) | —— | —— | 0.0157 (0.6300) | —— | −0.0461* (−1.7300) |
| Age | —— | —— | —— | —— | —— | −0.0466 (−1.6300) | −0.0427 (−1.5000) | −0.0434 (−1.5300) | −0.0501* (−1.7500) | −0.0524* (−1.8200) |
| Education | —— | —— | —— | —— | —— | 0.1842*** (5.9500) | 0.1902*** (6,1700) | 0.1898*** (6.1500) | 0.1851*** (5.9800) | 0.1819*** (5.8400) |
| Disease | —— | —— | —— | —— | —— | −0.4631*** (−15.8000) | −0.4519*** (−15.2800) | −0.4571*** (−15.7400) | −0.4643*** (−15.5600) | −0.4661*** (−15.5300) |
| Ln(GDP) | —— | —— | —— | —— | —— | 0.2744*** (8.6200) | 0.3667*** (12.3700) | 0.3297*** (9.8500) | 0.3104*** (9.1900) | 0.3131*** (9.1800) |
| Gender | —— | —— | —— | —— | —— | −0.1095* (−1.8700) | −0.0987* (−16900) | −0.0984* (−1.6900) | −0.1059* (−1.8100) | −0.1124* (−1.9100) |
| C | −1.1197 | 0.1106 | 0.0370 | −0.4554 | 1.2025 | 0.0770 | 0.6573 | 0.6432 | 0.6654 | 0.8187 |

Note: (1) We use mixed logistics regression to estimate Model (1). We use the ordinary likelihood ratio test to select a model that is better suited to current data analysis. (2) "z value" is enclosed in parentheses; *, **, and *** denote the levels of significance by 10%, 5%, and 1%, respectively. (3) We use coldaig2 to test the multicollinearity problems. Result for each regression are shown in Appendix C.1 Multicollinearity test for logistic regression. For those regressions which didn't pass the test, we conducted modification one by one.

## 6. Conclusions

Since the turn of the 21st century, environmental pollution has become a significant hindrance for developing countries when solving developmental problems. The major air pollution problem that China is facing is compound pollution of the atmosphere represented by PM2.5 and ozone. Many types of pollutants in the atmosphere are in high concentration levels, which are the root cause of frequent air pollution in major urban agglomerations in China. In recent years, haze, winds, and other extreme weather occur frequently, which affect the cityscape and people's daily lives. Specifically, respiratory and lung diseases have become the consequences of air pollution. In this study, 2011, 2013, and 2015 data in CHARLS database are used to analyze the impact of air pollution on physical health and medical insurance expenses empirically among the middle-aged and old Chinese people. The conclusions are as follows.

First, air pollutants, PM10, $SO_2$, and $NO_2$, have an impact on the health of the elderly people. Among these pollutants, the impact of PM10 shows an inverted U-shaped structure that initially increases and then decreases. That is, as PM10 concentration increases, the health condition initially becomes better and then worsens; whereas $SO_2$ does not show a stable and significant effect; $NO_2$ shows a downward trend, that is, the health status continues to deteriorate with the increase in concentration.

Second, health and medical insurance expenses are closely related; a worse self-rated health indicates more medical insurance expenses. If the number of patients suffering from chronic diseases increases, the total medical insurance expenditure will be reduced, and the total medical expenses will increase. Age, self-rated health, and local GDP significantly affect medical insurance expenses. The local economic development also has a direct relationship with the expenditure on medical insurance expenses of local residents. In developed areas, people have more channels to receive medical services with high quality, their own economic conditions can support the demand, and insurance is strong.

Finally, the concentration of air pollutant and self-rated health indicators have an impact on the medical insurance expenses of the elderly people. The severity of air pollution indirectly promotes the increase of medical insurance expenses by affecting the health conditions, and the impact on medical insurance expenses caused by different pollutants is also different. The results show that in the main air pollutants, only the changes of PM10 and $SO_2$ have a significant impact on medical insurance expenses, which shows a reverse relationship. In addition, significant differences are found in the effects of air pollution on the health status among populations affected by different diseases, such as hypertension and heart diseases.

Moreover, based on the robustness test, we found a difference in the types of major pollutants that influence health conditions between the north and south samples by dividing the city into two sub−samples of north and south with Qinling Ridge-Huaihe River as a boundary. The air quality level is worse in the north than that in the south due to burning coal for heating in the north. From the perspective of pollutant categories, PM10 accounts for a larger proportion in the north, whereas $NO_2$ is mainly present in the south. Moreover, $SO_2$ is the main pollutant hazardous to health in the north. This finding can provide some guidance to help us focus on the main target of air pollution control activity in the next stage.

The conclusion of this study verifies the practical significance of protecting the environment and purifying the air for the citizens and development of our country. Compared with the medical expenses caused by pure physical deterioration without air pollution, air pollution does affect people's health in real terms, and this impact can be reflected in people's medical insurance expenses. This paper presents seven regression models, which were constructed in such a way that each of them examines and verifies the relationship among three different objects: Air pollutants, self-rated health and medical insurance costs. All tested models were characterized and based on a progressive way, which indicates the interaction among air pollutants, self-rated health as well as medical insurance costs. The robustness test, where two different perspectives including adding outpatient frequency and hospitalization times, and distinguishing the difference between northern and southern China, support the result more solidly.

### 7. Discussion

Curbing air pollution can help maintain a suitable environmental quality and have a good effect on people's health, as well as reducing the costs for medical treatment, thereby decreasing social costs. This study is of an enlightening significance for improving air pollution, protecting people's physical and mental health, saving social medical costs, and implementing targeted air pollution control. This study also provides a useful perspective for the debate in China and the world about the deterioration of air pollution.

At the same time, due to the limited knowledge of the author and time constraints, this paper has some unsolved problems:

(1)   Further study on air quality difference between the north and south of China, and the difference in health, medical expenses and life expectancy caused by air pollution.

(2)   The concentrations of $SO_2$, $NO_2$ and PM10 are representative of air pollutants, but they are not comprehensive. For example, PM2.5, is small in particle size, rich in toxic substances, and has long residence time in the atmosphere and long transport distance. Therefore, the impact on human health and atmospheric environmental quality is greater, but PM2.5 monitoring indicators The data was only available in January 2013.

(3)   The data span is relatively short, the current CHARLS website only updated to 2015 data.

In view of the above problems, this paper puts forward the following ideas:

(1)   The coal-burning heating policy in the south and north of China is implemented with the Qinling-Huaihe River as the boundary, so we can consider the Regression Discontinuity with the latitude of the Qinling-Huaihe River as the breakpoint. The difference of air quality between the north and the south and its series influence are obtained.

(2)   Various important air pollutant concentrations can be incorporated into the calculation to construct a new "air quality composite index", similar to the air quality index "Air Quality Index" (AQI), which China began to monitor and publish in real time in May 2012.

(3)   To better measure the void, the national baseline survey data for 2017 will be updated in 2019, and the data will be updated to further verify the results of the article.

**Author Contributions:** All the authors have contributed to the whole development of the manuscript: designing the research, performing the calculations, writing the text, discussing the results, and obtaining the conclusions.

**Funding:** This paper is supported by the Key project of Chinese National Social Science Fund (Research on the Collaborative Path of the Evolution of Internet Lending and the Financing Advancement of Small and Micro Enterprises under the New Financial Normality, No. 15AJY018) and the key project of Chongqing University (No.CQDXWL-2014-Z019) and (No. 106112016CDJXY020013).

**Conflicts of Interest:** The authors declare no conflict of interest.

## Appendix A

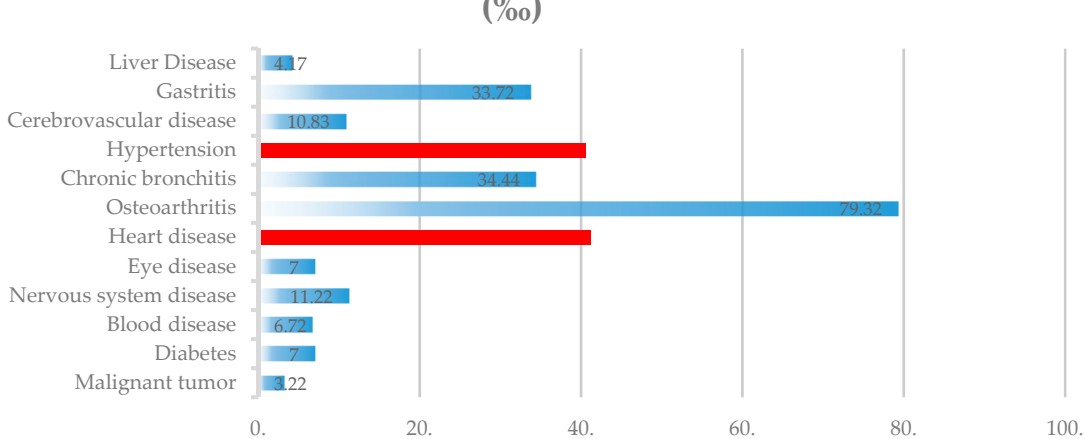

**Figure A1.** Prevalence of chronic diseases in the elderly. Note: The two lines on the red icon indicate that the prevalence of hypertension and heart disease is close. We selected the similar prevalence of heart disease and hypertension as a sub-sample basis to explore the different populations affected by air pollution.

## Appendix B

Take Table 4 as an example. After the regression, we used the White's (1980) test to test the heteroscedasticity:

**Table A1.** Heterogeneous test and correction for regression results before the presentation of Table 4.

| Variable | OLS Regression |
| --- | --- |
| Health | −110.5711 *** (−2.7600) |
| Ln GDP | −8.9002 (−0.3000) |
| Disease | 199.5100 *** (13.7300) |
| Education | 3.7920 (1.0300) |
| Gender | −44.1925 (−1.1400) |
| Age | −1.3731 (−0.7400) |
| C | 1016.7050 |

**Table A2.** Heteroscedasticity test.

| Source | Chi2 | df | p |
| --- | --- | --- | --- |
| Heteroscedasticity | 70.84 | 26 | 0.0000 |
| Skewness | 628.71 | 6 | 0.0000 |
| Kurtosis | 579.75 | 1 | 0.0000 |
| Total | 1279.31 | 33 | 0.0000 |

White's test for H0: Homoscedasticity

Against Ha: Unrestricted heteroscedasticity

Chi2 (26) = 70.84

Prob > chi2 = 0.0000

From the above regression results, Prob > Chi2 = 0.0000, significantly less than 0.05, indicating that the null hypothesis of "no heteroscedasticity" is rejected, indicating that there is heteroscedasticity. Next, the weighted least squares method (WLS) is used to correct the heteroscedasticity. The corrected regression results are shown in Table 4 in the main text.

In this paper, we have tested the heteroscedasticity for each OLS regression. The results of heteroscedasticity in the test are as follows (Columns 1, 2, and 3 of Table 5), and the test results without heteroscedasticity are not listed

**Table A3.** Heteroscedasticity test.

| Source | Chi2 | df | p |
|---|---|---|---|
| Heteroscedasticity | 132.29 | 52 | 0.0000 |
| Skewness | 631.30 | 9 | 0.0000 |
| Kurtosis | 570.24 | 1 | 0.0000 |
| Total | 1333.84 | 62 | 0.0000 |

Note: Table 5 results of heteroscedasticity test in the first column of regression.

White's test for H0: Homoscedasticity
Against Ha: Unrestricted heteroscedasticity
Chi2 (52) = 132.29
Prob > chi2 = 0.0000

**Table A4.** Heteroscedasticity test.

| Source | Chi2 | df | p |
|---|---|---|---|
| Heteroscedasticity | 114.01 | 49 | 0.0000 |
| Skewness | 624.85 | 9 | 0.0000 |
| Kurtosis | 570.50 | 1 | 0.0000 |
| Total | 1309.36 | 59 | 0.0000 |

Note: Table 5 results of heteroscedasticity test in the second column of regression.

White's test for H0: Homoscedasticity
Against Ha: Unrestricted heteroscedasticity
Chi2 (49) = 114.01
Prob > chi2 = 0.0000

**Table A5.** Heteroscedasticity test.

| Source | Chi2 | df | p |
|---|---|---|---|
| Heteroscedasticity | 142.43 | 73 | 0.0000 |
| Skewness | 633.56 | 12 | 0.0000 |
| Kurtosis | 570.34 | 1 | 0.0000 |
| Total | 1346.32 | 86 | 0.0000 |

Note: Table 5 results of heteroscedasticity test in the third column of regression.

White's test for H0: homoscedasticity
Against Ha: unrestricted heteroscedasticity
Chi2 (73) = 142.43 s
Prob > chi2 = 0.0000

All the above regression with heteroscedasticity are corrected by the weighted least squares method (WLS), and is displayed in the corresponding position in the text.

## Appendix C

For both logistic regression models and OLS regression models, we added a multicollinearity test. The final results were shown below.

*Appendix C.1. Multicollinearity Test for Logistic Regression*

We used coldaig2 to test the multicollinearity problems. Coldiag 2 first computes the condition number of the matrix. If this number is "large" (Belsley et al. suggest 30 or higher), then there may be collinearity problems. All rest results after modification are shown below.

**Table A6.** Multicollinearity test.

|   | Index | _cons | PM10 | PM10$^2$ |
|---|-------|-------|------|----------|
| 1 | 1.00 | 0.01 | 0.00 | 0.01 |
| 2 | 2.82 | 0.06 | 0.00 | 0.08 |
| 3 | 16.37 | 0.93 | 1.00 | 0.91 |

Coldiag 2
Condition number using scaled variables = 16.37
Condition Indexes and Variance-Decomposition Proportions
Condition

**Table A7.** Multicollinearity test.

|   | Index | _cons | PM10 | PM10$^2$ |
|---|-------|-------|------|----------|
| 1 | 1.00 | 0.01 | 0.02 | 0.02 |
| 2 | 1.42 | 0.99 | 0.00 | 0.00 |
| 3 | 7.86 | 0.01 | 0.98 | 0.98 |

Coldiag2
Condition number using scaled variables = 7.86
Condition Indexes and Variance-Decomposition Proportions
Condition

**Table A8.** Multicollinearity test.

|   | Index | _cons | PM10 | PM10$^2$ |
|---|-------|-------|------|----------|
| 1 | 1.00 | 0.17 | 0.02 | 0.18 |
| 2 | 1.28 | 0.04 | 0.90 | 0.00 |
| 3 | 2.14 | 0.79 | 0.08 | 0.82 |

Coldiag2
Condition number using scaled variables = 2.14
Condition Indexes and Variance-Decomposition Proportions
Condition

**Table A9.** Multicollinearity test.

|   | Index | _cons | PM10 | PM10$^2$ |
|---|-------|-------|------|----------|
| 1 | 1.00 | 0.01 | 0.01 | 0.01 |
| 2 | 1.72 | 0.00 | 0.00 | 0.00 |
| 3 | 5.84 | 0.53 | 0.64 | 0.00 |
| 4 | 0.39 | 0.46 | 0.35 | 0.99 |

Coldiag2
Condition number using scaled variables = 8.39
Condition Indexes and Variance-Decomposition Proportions
Condition

**Table A10.** Multicollinearity test.

| | Index | _cons | PM10 | PM10$^2$ | NO$_2$ | NO$_2$$^2$ | SO$_2$ | SO$_2$$^2$ |
|---|---|---|---|---|---|---|---|---|
| 1 | 1.00 | 0.02 | 0.02 | 0.03 | 0.02 | 0.04 | 0.00 | 0.00 |
| 2 | 1.20 | 0.01 | 0.02 | 0.01 | 0.01 | 0.00 | 0.01 | 0.01 |
| 3 | 1.33 | 0.15 | 0.04 | 0.08 | 0.08 | 0.08 | 0.00 | 0.00 |
| 4 | 1.94 | 0.04 | 0.01 | 0.38 | 0.38 | 0.38 | 0.00 | 0.00 |
| 5 | 2.70 | 0.63 | 0.00 | 0.03 | 0.03 | 0.03 | 0.00 | 0.00 |
| 6 | 3.80 | 0.15 | 0.91 | 0.47 | 0.47 | 0.47 | 0.00 | 0.00 |
| 7 | 9.06 | 0.00 | 0.00 | 0.00 | 0.00 | 0.00 | 0.98 | 0.98 |

Coldiag2
Condition number using scaled variables = 9.06
Condition Indexes and Variance-Decomposition Proportions
Condition

**Table A11.** Multicollinearity test.

| | Index | _cons | PM10 | PM10$^2$ | NO$_2$ | NO$_2$$^2$ | SO$_2$ | SO$_2$$^2$ |
|---|---|---|---|---|---|---|---|---|
| 1 | 1.00 | 0.02 | 0.02 | 0.03 | 0.02 | 0.04 | 0.00 | 0.00 |
| 2 | 1.18 | 0.01 | 0.01 | 0.03 | 0.01 | 0.01 | 0.01 | 0.01 |
| 3 | 1.32 | 0.13 | 0.05 | 0.00 | 0.08 | 0.07 | 0.00 | 0.00 |
| 4 | 1.92 | 0.04 | 0.01 | 0.24 | 0.38 | 0.04 | 0.00 | 0.00 |
| 5 | 2.68 | 0.59 | 0.00 | 0.02 | 0.03 | 0.82 | 0.00 | 0.00 |
| 6 | 3.77 | 0.14 | 0.91 | 0.69 | 0.47 | 0.00 | 0.00 | 0.00 |
| 7 | 8.88 | 0.07 | 0.00 | 0.00 | 0.00 | 0.98 | 0.98 | 0.98 |

Coldiag2
Condition number using scaled variables = 8.88
Condition Indexes and Variance-Decomposition Proportions
Condition

**Table A12.** Multicollinearity test.

| | Index | _cons | PM10 | PM10$^2$ | Age | Education | Disease | lngdp | Gender |
|---|---|---|---|---|---|---|---|---|---|
| 1 | 1.00 | 0.06 | 0.03 | 0.06 | 0.00 | 0.00 | 0.00 | 0.01 | 0.06 |
| 2 | 1.14 | 0.04 | 0.10 | 0.03 | 0.00 | 0.04 | 0.01 | 0.07 | 0.06 |
| 3 | 1.36 | 0.00 | 0.00 | 0.02 | 0.30 | 0.00 | 0.27 | 0.23 | 0.00 |
| 4 | 1.38 | 0.01 | 0.01 | 0.01 | 0.29 | 0.47 | 0.05 | 0.03 | 0.00 |
| 5 | 1.47 | 0.01 | 0.02 | 0.02 | 0.13 | 0.10 | 0.63 | 0.05 | 0.00 |
| 6 | 1.57 | 0.00 | 0.00 | 0.04 | 0.27 | 0.30 | 0.04 | 0.40 | 0.01 |
| 7 | 2.63 | 0.22 | 0.36 | 0.31 | 0.01 | 0.05 | 0.00 | 0.13 | 0.53 |
| 8 | 3.07 | 0.65 | 0.47 | 0.52 | 0.00 | 0.03 | 0.00 | 0.09 | 0.34 |

Coldiag2
Condition number using scaled variables = 3.07
Condition Indexes and Variance-Decomposition Proportions
Condition

**Table A13.** Multicollinearity test.

| | Index | _cons | SO$_2$ | SO$_2$$^2$ | Age | Education | Disease | lngdp | Gender |
|---|---|---|---|---|---|---|---|---|---|
| 1 | 1.00 | 0.01 | 0.02 | 0.02 | 0.00 | 0.00 | 0.00 | 0.00 | 0.01 |
| 2 | 1.07 | 0.12 | 0.00 | 0.00 | 0.00 | 0.00 | 0.00 | 0.00 | 0.12 |
| 3 | 1.33 | 0.01 | 0.00 | 0.02 | 0.30 | 0.03 | 0.04 | 0.47 | 0.00 |
| 4 | 1.36 | 0.00 | 0.00 | 0.00 | 0.29 | 0.55 | 0.09 | 0.02 | 0.00 |
| 5 | 1.42 | 0.00 | 0.00 | 0.00 | 0.13 | 0.11 | 0.87 | 0.03 | 0.00 |
| 6 | 1.57 | 0.01 | 0.00 | 0.00 | 0.27 | 0.30 | 0.00 | 0.47 | 0.00 |
| 7 | 2.79 | 0.83 | 0.00 | 0.00 | 0.01 | 0.01 | 0.00 | 0.00 | 0.87 |
| 8 | 7.79 | 0.03 | 0.98 | 0.98 | 0.00 | 0.00 | 0.00 | 0.00 | 0.00 |

Coldiag2
Condition number using scaled variables = 7.79
Condition Indexes and Variance-Decomposition Proportions
Condition

**Table A14.** Multicollinearity test.

| | Index | _cons | NO$_2$ | NO$_2$$^2$ | Age | Education | Disease | Lngdp | Gender |
|---|---|---|---|---|---|---|---|---|---|
| 1 | 1.00 | 0.06 | 0.00 | 0.07 | 0.00 | 0.00 | 0.00 | 0.00 | 0.07 |
| 2 | 1.21 | 0.00 | 0.22 | 0.00 | 0.00 | 0.05 | 0.00 | 0.23 | 0.00 |
| 3 | 1.44 | 0.00 | 0.00 | 0.00 | 0.52 | 0.25 | 0.10 | 0.02 | 0.00 |
| 4 | 1.48 | 0.00 | 0.01 | 0.00 | 0.02 | 0.12 | 0.78 | 0.01 | 0.00 |
| 5 | 1.58 | 0.01 | 0.06 | 0.00 | 0.41 | 0.49 | 0.04 | 0.01 | 0.00 |
| 6 | 1.92 | 0.01 | 0.21 | 0.32 | 0.01 | 0.00 | 0.07 | 0.35 | 0.13 |
| 7 | 2.20 | 0.00 | 0.47 | 0.35 | 0.03 | 0.03 | 0.00 | 0.38 | 0.17 |
| 8 | 3.20 | 0.92 | 0.02 | 0.25 | 0.00 | 0.07 | 0.00 | 0.00 | 0.63 |

Coldiag2
Condition number using scaled variables = 3.20
Condition Indexes and Variance-Decomposition Proportions
Condition

**Table A15.** Multicollinearity test.

| | Index | _cons | PM10 | PM10$^2$ | SO$_2$ | SO$_2$$^2$ | NO$_2$ | NO$_2$$^2$ | Age | Education | Disease | Lngdp | Gender |
|---|---|---|---|---|---|---|---|---|---|---|---|---|---|
| 1 | 1.00 | 0.02 | 0.01 | 0.02 | 0.00 | 0.00 | 0.02 | 0.04 | 0.00 | 0.00 | 0.00 | 0.01 | 0.02 |
| 2 | 1.17 | 0.03 | 0.01 | 0.00 | 0.00 | 0.00 | 0.03 | 0.02 | 0.00 | 0.01 | 0.00 | 0.03 | 0.04 |
| 3 | 1.24 | 0.00 | 0.02 | 0.02 | 0.01 | 0.01 | 0.02 | 0.00 | 0.00 | 0.01 | 0.00 | 0.03 | 0.00 |
| 4 | 1.57 | 0.01 | 0.01 | 0.04 | 0.00 | 0.00 | 0.01 | 0.00 | 0.20 | 0.00 | 0.26 | 0.18 | 0.01 |
| 5 | 1.61 | 0.01 | 0.01 | 0.01 | 0.00 | 0.00 | 0.00 | 0.00 | 0.33 | 0.45 | 0.03 | 0.02 | 0.00 |
| 6 | 1.72 | 0.01 | 0.01 | 0.02 | 0.00 | 0.00 | 0.00 | 0.00 | 0.15 | 0.11 | 0.61 | 0.04 | 0.00 |
| 7 | 1.77 | 0.00 | 0.00 | 0.05 | 0.00 | 0.00 | 0.04 | 0.00 | 0.29 | 0.32 | 0.08 | 0.10 | 0.02 |
| 8 | 1.35 | 0.00 | 0.00 | 0.10 | 0.00 | 0.00 | 0.17 | 0.45 | 0.00 | 0.00 | 0.01 | 0.16 | 0.12 |
| 9 | 2.43 | 0.00 | 0.01 | 0.04 | 0.00 | 0.00 | 0.27 | 0.21 | 0.04 | 0.04 | 0.00 | 0.44 | 0.17 |
| 10 | 3.47 | 0.53 | 0.14 | 0.12 | 0.00 | 0.00 | 0.14 | 0.24 | 0.00 | 0.05 | 0.00 | 0.00 | 0.50 |
| 11 | 4.05 | 0.35 | 0.77 | 0.58 | 0.00 | 0.00 | 0.29 | 0.02 | 0.00 | 0.01 | 0.00 | 0.00 | 0.11 |
| 12 | 9.33 | 0.05 | 0.00 | 0.00 | 0.98 | 0.98 | 0.00 | 0.02 | 0.00 | 0.00 | 0.00 | 0.00 | 0.00 |

Coldiag2
Condition number using scaled variables = 9.33
Condition Indexes and Variance-Decomposition Proportions
Condition

**Table A16.** Multicollinearity test.

| | Index | _cons | Age | Education | Disease | lngdp | Gender |
|---|---|---|---|---|---|---|---|
| 1 | 1.00 | 0.12 | 0.00 | 0.01 | 0.00 | 0.00 | 0.13 |
| 2 | 1.25 | 0.01 | 0.02 | 0.43 | 0.01 | 0.38 | 0.00 |
| 3 | 1.28 | 0.00 | 0.61 | 0.02 | 0.15 | 0.14 | 0.00 |
| 4 | 1.33 | 0.00 | 0.08 | 0.05 | 0.82 | 0.05 | 0.00 |
| 5 | 1.46 | 0.01 | 0.28 | 0.41 | 0.03 | 0.42 | 0.00 |
| 6 | 2.64 | 0.86 | 0.01 | 0.07 | 0.00 | 0.00 | 0.87 |

Coldiag2
Condition number using scaled variables = 2.64
Condition Indexes and Variance-Decomposition Proportions
Condition

**Table A17.** Multicollinearity test.

| | Index | _cons | Age | Education | Disease | lngdp | Gender |
|---|---|---|---|---|---|---|---|
| 1 | 1.00 | 0.12 | 0.00 | 0.00 | 0.00 | 0.00 | 0.12 |
| 2 | 1.20 | 0.00 | 0.01 | 0.35 | 0.03 | 0.38 | 0.00 |
| 3 | 1.31 | 0.00 | 0.46 | 0.02 | 0.02 | 0.00 | 0.00 |
| 4 | 1.35 | 0.00 | 0.52 | 0.05 | 0.05 | 0.01 | 0.00 |
| 5 | 1.50 | 0.00 | 0.00 | 0.55 | 0.55 | 0.61 | 0.00 |
| 6 | 2.73 | 0.87 | 0.01 | 0.03 | 0.03 | 0.00 | 0.88 |

Coldiag2
Condition number using scaled variables = 2.73
Condition Indexes and Variance-Decomposition Proportions
Condition

**Table A18.** Multicollinearity test.

| | Index | _cons | PM10 | PM10$^2$ | SO$_2$ | SO$_2{}^2$ | NO$_2$ | NO$_2{}^2$ |
|---|---|---|---|---|---|---|---|---|
| 1 | 1.00 | 0.02 | 0.03 | 0.05 | 0.00 | 0.00 | 0.03 | 0.04 |
| 2 | 1.12 | 0.02 | 0.01 | 0.03 | 0.01 | 0.01 | 0.01 | 0.02 |
| 3 | 1.27 | 0.13 | 0.09 | 0.00 | 0.00 | 0.00 | 0.12 | 0.07 |
| 4 | 1.83 | 0.06 | 0.03 | 0.27 | 0.00 | 0.00 | 0.55 | 0.02 |
| 5 | 2.53 | 0.62 | 0.00 | 0.06 | 0.00 | 0.00 | 0.01 | 0.85 |
| 6 | 2.75 | 0.12 | 0.83 | 0.59 | 0.00 | 0.00 | 0.28 | 0.00 |
| 7 | 11.91 | 0.03 | 0.00 | 0.00 | 0.99 | 0.99 | 0.00 | 0.01 |

Coldiag2
Condition number using scaled variables = 11.91
Condition Indexes and Variance-Decomposition Proportions
Condition

**Table A19.** Multicollinearity test.

| | Index | _cons | PM10 | PM10$^2$ | SO$_2$ | SO$_2{}^2$ | NO$_2$ | NO$_2{}^2$ |
|---|---|---|---|---|---|---|---|---|
| 1 | 1.00 | 0.00 | 0.02 | 0.03 | 0.00 | 0.00 | 0.02 | 0.03 |
| 2 | 1.09 | 0.00 | 0.02 | 0.04 | 0.00 | 0.00 | 0.01 | 0.02 |
| 3 | 1.21 | 0.02 | 0.05 | 0.00 | 0.00 | 0.00 | 0.08 | 0.09 |
| 4 | 1.65 | 0.00 | 0.01 | 0.23 | 0.00 | 0.00 | 0.44 | 0.03 |
| 5 | 2.52 | 0.13 | 0.04 | 0.03 | 0.00 | 0.00 | 0.06 | 0.76 |
| 6 | 3.12 | 0.05 | 0.71 | 0.65 | 0.00 | 0.00 | 0.36 | 0.04 |
| 7 | 169.10 | 0.79 | 0.15 | 0.02 | 1.00 | 1.00 | 0.02 | 0.03 |

Coldiag2
Condition number using scaled variables = 169.10
Condition Indexes and Variance-Decomposition Proportions
Condition

**Table A20.** Multicollinearity test.

|   | Index | _cons | PM10 | PM10$^2$ |
|---|-------|-------|------|----------|
| 1 | 1.00  | 0.00  | 0.00 | 0.01     |
| 2 | 2.96  | 0.06  | 0.00 | 0.09     |
| 3 | 17.77 | 0.94  | 1.00 | 0.91     |

Coldiag2
Condition number using scaled variables = 17.77
Condition Indexes and Variance-Decomposition Proportions
Condition

**Table A21.** Multicollinearity test.

|   | Index | _cons | PM10 | PM10$^2$ |
|---|-------|-------|------|----------|
| 1 | 1.00  | 0.01  | 0.02 | 0.02     |
| 2 | 1.42  | 0.98  | 0.00 | 0.00     |
| 3 | 7.88  | 0.01  | 0.98 | 0.98     |

Coldiag2
Condition number using scaled variables = 7.86
Condition Indexes and Variance-Decomposition Proportions
Condition

**Table A22.** Multicollinearity test.

|   | Index | _cons | PM10 | PM10$^2$ |
|---|-------|-------|------|----------|
| 1 | 1.00  | 0.00  | 0.00 | 0.00     |
| 2 | 3.75  | 0.06  | 0.00 | 0.04     |
| 3 | 28.63 | 0.94  | 1.00 | 0.96     |

Coldiag2
Condition number using scaled variables = 28.63
Condition Indexes and Variance-Decomposition Proportions
Condition

**Table A23.** Multicollinearity test.

|   | Index | _cons | PM10 | PM10$^2$ |
|---|-------|-------|------|----------|
| 1 | 1.00  | 0.01  | 0.01 | 0.01     |
| 2 | 1.74  | 0.00  | 0.00 | 0.00     |
| 3 | 6.35  | 0.88  | 0.41 | 0.04     |
| 4 | 0.39  | 0.46  | 0.35 | 0.99     |

Coldiag2
Condition number using scaled variables = 7.75
Condition Indexes and Variance-Decomposition Proportions
Condition

**Table A24.** Multicollinearity test.

|   | Index | _cons | PM10 | PM10$^2$ | NO$_2$ | NO$_2$$^2$ | SO$_2$ | SO$_2$$^2$ |
|---|-------|-------|------|----------|--------|------------|--------|------------|
| 1 | 1.00 | 0.01 | 0.02 | 0.02 | 0.02 | 0.02 | 0.00 | 0.00 |
| 2 | 1.32 | 0.02 | 0.01 | 0.01 | 0.02 | 0.01 | 0.01 | 0.01 |
| 3 | 1.54 | 0.19 | 0.06 | 0.06 | 0.00 | 0.03 | 0.00 | 0.00 |
| 4 | 2.38 | 0.14 | 0.03 | 0.03 | 0.19 | 0.00 | 0.00 | 0.00 |
| 5 | 3.22 | 0.45 | 0.54 | 0.54 | 0.03 | 0.33 | 0.00 | 0.00 |
| 6 | 3.89 | 0.07 | 0.34 | 0.34 | 0.71 | 0.57 | 0.00 | 0.00 |
| 7 | 9.75 | 0.12 | 0.00 | 0.00 | 0.01 | 0.04 | 0.98 | 0.98 |

Coldiag2
Condition number using scaled variables = 9.75
Condition Indexes and Variance-Decomposition Proportions
Condition

**Table A25.** Multicollinearity test.

|   | Index | _cons | PM10 | PM10$^2$ | Age | Education | Disease | lngdp |
|---|-------|-------|------|----------|-----|-----------|---------|-------|
| 1 | 1.00 | 0.06 | 0.03 | 0.06 | 0.06 | 0.00 | 0.00 | 0.00 |
| 2 | 1.14 | 0.03 | 0.11 | 0.03 | 0.03 | 0.03 | 0.03 | 0.05 |
| 3 | 1.33 | 0.02 | 0.00 | 0.03 | 0.03 | 0.24 | 0.08 | 0.32 |
| 4 | 1.36 | 0.00 | 0.00 | 0.00 | 0.00 | 0.20 | 0.03 | 0.03 |
| 5 | 1.45 | 0.00 | 0.02 | 0.02 | 0.02 | 0.00 | 0.86 | 0.02 |
| 6 | 1.57 | 0.01 | 0.00 | 0.02 | 0.02 | 0.40 | 0.00 | 0.27 |
| 7 | 2.66 | 0.25 | 0.34 | 0.28 | 0.28 | 0.08 | 0.00 | 0.16 |
| 8 | 3.80 | 0.68 | 0.40 | 0.55 | 0.55 | 0.03 | 0.00 | 0.12 |

Coldiag2
Condition number using scaled variables = 3.09
Condition Indexes and Variance-Decomposition Proportions
Condition

**Table A26.** Multicollinearity test.

|   | Index | _cons | SO$_2$ | SO$_2$$^2$ | Age | Education | Disease | lngdp | Gender |
|---|-------|-------|--------|------------|-----|-----------|---------|-------|--------|
| 1 | 1.00 | 0.01 | 0.01 | 0.02 | 0.00 | 0.00 | 0.00 | 0.00 | 0.01 |
| 2 | 1.07 | 0.11 | 0.00 | 0.00 | 0.00 | 0.01 | 0.00 | 0.00 | 0.12 |
| 3 | 1.32 | 0.01 | 0.00 | 0.00 | 0.01 | 0.01 | 0.00 | 0.37 | 0.00 |
| 4 | 1.36 | 0.00 | 0.00 | 0.00 | 0.64 | 0.64 | 0.06 | 0.14 | 0.00 |
| 5 | 1.42 | 0.00 | 0.00 | 0.00 | 0.04 | 0.04 | 0.93 | 0.01 | 0.00 |
| 6 | 1.60 | 0.02 | 0.00 | 0.00 | 0.30 | 0.30 | 0.00 | 0.46 | 0.00 |
| 7 | 2.82 | 0.80 | 0.00 | 0.00 | 0.01 | 0.01 | 0.00 | 0.01 | 0.07 |
| 8 | 7.76 | 0.05 | 0.98 | 0.98 | 0.00 | 0.00 | 0.00 | 0.00 | 0.00 |

Coldiag2
Condition number using scaled variables = 7.76
Condition Indexes and Variance-Decomposition Proportions
Condition

**Table A27.** Multicollinearity test.

|  | Index | _cons | $NO_2$ | $NO_2{}^2$ | Age | Education | Disease | lngdp | Gender |
|---|---|---|---|---|---|---|---|---|---|
| 1 | 1.00 | 0.06 | 0.01 | 0.07 | 0.00 | 0.00 | 0.00 | 0.00 | 0.06 |
| 2 | 1.19 | 0.01 | 0.16 | 0.01 | 0.00 | 0.05 | 0.01 | 0.18 | 0.02 |
| 3 | 1.42 | 0.00 | 0.00 | 0.00 | 0.55 | 0.26 | 0.03 | 0.01 | 0.00 |
| 4 | 1.47 | 0.01 | 0.00 | 0.00 | 0.00 | 0.10 | 0.78 | 0.03 | 0.00 |
| 5 | 1.60 | 0.02 | 0.07 | 0.00 | 0.39 | 0.44 | 0.10 | 0.01 | 0.00 |
| 6 | 1.82 | 0.01 | 0.09 | 0.23 | 0.02 | 0.01 | 0.07 | 0.45 | 0.09 |
| 7 | 2.43 | 0.00 | 0.54 | 0.38 | 0.02 | 0.05 | 0.00 | 0.31 | 0.26 |
| 8 | 3.25 | 0.90 | 0.12 | 0.32 | 0.01 | 0.08 | 0.00 | 0.01 | 0.57 |

Coldiag2
Condition number using scaled variables = 3.25
Condition Indexes and Variance-Decomposition Proportions
Condition

**Table A28.** Multicollinearity test.

|  | Index | _cons | PM10 | $SO_2$ | $NO_2$ | Age | Education | Disease | Lngdp | Gender |
|---|---|---|---|---|---|---|---|---|---|---|
| 1 | 1.00 | 0.00 | 0.09 | 0.04 | 0.09 | 0.00 | 0.01 | 0.01 | 0.08 | 0.00 |
| 2 | 1.08 | 0.12 | 0.00 | 0.00 | 0.00 | 0.00 | 0.01 | 0.00 | 0.00 | 0.13 |
| 3 | 1.36 | 0.01 | 0.01 | 0.05 | 0.00 | 0.28 | 0.46 | 0.01 | 0.02 | 0.00 |
| 4 | 1.38 | 0.00 | 0.00 | 0.12 | 0.00 | 0.29 | 0.01 | 0.32 | 0.12 | 0.00 |
| 5 | 1.47 | 0.00 | 0.00 | 0.22 | 0.00 | 0.13 | 0.02 | 0.63 | 0.01 | 0.00 |
| 6 | 1.55 | 0.01 | 0.03 | 0.47 | 0.02 | 0.22 | 0.28 | 0.01 | 0.01 | 0.00 |
| 7 | 1.75 | 0.01 | 0.25 | 0.09 | 0.01 | 0.06 | 0.10 | 0.03 | 0.57 | 0.00 |
| 8 | 2.45 | 0.00 | 0.61 | 0.01 | 0.87 | 0.00 | 0.00 | 0.00 | 0.18 | 0.00 |
| 9 | 2.85 | 0.85 | 0.00 | 0.00 | 0.00 | 0.01 | 0.10 | 0.00 | 0.01 | 0.87 |

Coldiag2
Condition number using scaled variables = 2.85
Condition Indexes and Variance-Decomposition Proportions
Condition

**Table A29.** Multicollinearity test.

|  | Index | _cons | PM10 | $PM10^2$ | $SO_2$ | $SO_2{}^2$ | $NO_2$ | $NO_2{}^2$ | Age | Education | Disease | Lngdp | Gender |
|---|---|---|---|---|---|---|---|---|---|---|---|---|---|
| 1 | 1.00 | 0.01 | 0.02 | 0.02 | 0.00 | 0.00 | 0.02 | 0.02 | 0.00 | 0.00 | 0.00 | 0.00 | 0.01 |
| 2 | 1.27 | 0.04 | 0.01 | 0.00 | 0.00 | 0.00 | 0.02 | 0.01 | 0.00 | 0.01 | 0.00 | 0.02 | 0.06 |
| 3 | 1.39 | 0.00 | 0.03 | 0.01 | 0.01 | 0.01 | 0.03 | 0.00 | 0.00 | 0.01 | 0.00 | 0.03 | 0.01 |
| 4 | 1.66 | 0.02 | 0.01 | 0.04 | 0.00 | 0.00 | 0.01 | 0.00 | 0.04 | 0.08 | 0.11 | 0.23 | 0.02 |
| 5 | 1.71 | 0.00 | 0.00 | 0.00 | 0.00 | 0.00 | 0.00 | 0.00 | 0.50 | 0.33 | 0.01 | 0.00 | 0.00 |
| 6 | 1.83 | 0.00 | 0.01 | 0.01 | 0.00 | 0.00 | 0.00 | 0.00 | 0.05 | 0.00 | 0.86 | 0.03 | 0.01 |
| 7 | 1.93 | 0.00 | 0.00 | 0.02 | 0.00 | 0.00 | 0.02 | 0.01 | 0.38 | 0.43 | 0.00 | 0.09 | 0.02 |
| 8 | 2.85 | 0.00 | 0.00 | 0.01 | 0.00 | 0.00 | 0.18 | 0.24 | 0.02 | 0.06 | 0.00 | 0.28 | 0.30 |
| 9 | 2.99 | 0.01 | 0.27 | 0.03 | 0.00 | 0.00 | 0.67 | 0.07 | 0.00 | 0.00 | 0.00 | 0.19 | 0.05 |
| 10 | 3.90 | 0.68 | 0.31 | 0.18 | 0.00 | 0.00 | 0.00 | 0.03 | 0.01 | 0.05 | 0.00 | 0.05 | 0.45 |
| 11 | 4.17 | 0.16 | 0.26 | 0.66 | 0.00 | 0.00 | 0.05 | 0.58 | 0.00 | 0.02 | 0.00 | 0.08 | 0.07 |
| 12 | 10.07 | 0.08 | 0.01 | 0.02 | 0.98 | 0.98 | 0.00 | 0.04 | 0.00 | 0.00 | 0.00 | 0.00 | 0.00 |

Coldiag2
Condition number using scaled variables = 10.07
Condition Indexes and Variance-Decomposition Proportions
Condition

**Table A30.** Multicollinearity test.

|   | Index | _cons | PM10 | PM10$^2$ |
|---|-------|-------|------|----------|
| 1 | 1.00 | 0.00 | 0.00 | 0.01 |
| 2 | 3.03 | 0.06 | 0.00 | 0.09 |
| 3 | 19.68 | 0.94 | 1.00 | 0.93 |

Coldiag2
Condition number using scaled variables = 19.68
Condition Indexes and Variance-Decomposition Proportions
Condition

**Table A31.** Multicollinearity test.

|   | Index | _cons | PM10 | PM10$^2$ |
|---|-------|-------|------|----------|
| 1 | 1.00 | 0.01 | 0.02 | 0.02 |
| 2 | 1.42 | 0.98 | 0.00 | 0.00 |
| 3 | 7.88 | 0.01 | 0.98 | 0.98 |

Coldiag2
Condition number using scaled variables = 17.70
Condition Indexes and Variance-Decomposition Proportions
Condition

**Table A32.** Multicollinearity test.

|   | Index | _cons | PM10 | PM10$^2$ |
|---|-------|-------|------|----------|
| 1 | 1.00 | 0.00 | 0.00 | 0.00 |
| 2 | 3.75 | 0.06 | 0.00 | 0.04 |
| 3 | 28.63 | 0.94 | 1.00 | 0.96 |

Coldiag2
Condition number using scaled variables = 2.19
Condition Indexes and Variance-Decomposition Proportions
Condition

**Table A33.** Multicollinearity test.

|   | Index | _cons | PM10 | PM10$^2$ |
|---|-------|-------|------|----------|
| 1 | 1.00 | 0.01 | 0.01 | 0.01 |
| 2 | 1.74 | 0.00 | 0.00 | 0.00 |
| 3 | 6.35 | 0.88 | 0.41 | 0.04 |
| 4 | 0.39 | 0.46 | 0.35 | 0.99 |

Coldiag2
Condition number using scaled variables = 11.42
Condition Indexes and Variance-Decomposition Proportions
Condition

**Table A34.** Multicollinearity test.

|   | Index | _cons | PM10 | PM10$^2$ | NO$_2$ | NO$_2$$^2$ | SO$_2$ | SO$_2$$^2$ |
|---|-------|-------|------|----------|--------|-----------|--------|-----------|
| 1 | 1.00 | 0.01 | 0.02 | 0.02 | 0.02 | 0.02 | 0.00 | 0.00 |
| 2 | 1.32 | 0.02 | 0.01 | 0.01 | 0.02 | 0.01 | 0.01 | 0.01 |
| 3 | 1.54 | 0.19 | 0.06 | 0.06 | 0.00 | 0.03 | 0.00 | 0.00 |
| 4 | 2.38 | 0.14 | 0.03 | 0.03 | 0.19 | 0.00 | 0.00 | 0.00 |
| 5 | 3.22 | 0.45 | 0.54 | 0.54 | 0.03 | 0.33 | 0.00 | 0.00 |
| 6 | 3.89 | 0.07 | 0.34 | 0.34 | 0.71 | 0.57 | 0.00 | 0.00 |
| 7 | 9.75 | 0.12 | 0.00 | 0.00 | 0.01 | 0.04 | 0.98 | 0.98 |

Coldiag2
Condition number using scaled variables = 30.01
Condition Indexes and Variance-Decomposition Proportions
Condition

**Table A35.** Multicollinearity test.

|   | Index | _cons | PM10 | PM10$^2$ | Age | Education | Disease | Lngdp |
|---|-------|-------|------|----------|-----|-----------|---------|-------|
| 1 | 1.00 | 0.06 | 0.03 | 0.06 | 0.06 | 0.00 | 0.00 | 0.00 |
| 2 | 1.14 | 0.03 | 0.11 | 0.03 | 0.03 | 0.03 | 0.03 | 0.05 |
| 3 | 1.33 | 0.02 | 0.00 | 0.03 | 0.03 | 0.24 | 0.08 | 0.32 |
| 4 | 1.36 | 0.00 | 0.00 | 0.00 | 0.00 | 0.20 | 0.03 | 0.03 |
| 5 | 1.45 | 0.00 | 0.02 | 0.02 | 0.02 | 0.00 | 0.86 | 0.02 |
| 6 | 1.57 | 0.01 | 0.00 | 0.02 | 0.02 | 0.40 | 0.00 | 0.27 |
| 7 | 2.66 | 0.25 | 0.34 | 0.28 | 0.28 | 0.08 | 0.00 | 0.16 |
| 8 | 3.80 | 0.68 | 0.40 | 0.55 | 0.55 | 0.03 | 0.00 | 0.12 |

Coldiag2
Condition number using scaled variables = 3.25
Condition Indexes and Variance-Decomposition Proportions
Condition

**Table A36.** Multicollinearity test.

|   | Index | _cons | SO$_2$ | SO$_2$$^2$ | Age | Education | Disease | Lngdp | Gender |
|---|-------|-------|--------|-----------|-----|-----------|---------|-------|--------|
| 1 | 1.00 | 0.01 | 0.01 | 0.02 | 0.00 | 0.00 | 0.00 | 0.00 | 0.01 |
| 2 | 1.07 | 0.11 | 0.00 | 0.00 | 0.00 | 0.01 | 0.00 | 0.00 | 0.12 |
| 3 | 1.32 | 0.01 | 0.00 | 0.00 | 0.01 | 0.01 | 0.00 | 0.37 | 0.00 |
| 4 | 1.36 | 0.00 | 0.00 | 0.00 | 0.64 | 0.64 | 0.06 | 0.14 | 0.00 |
| 5 | 1.42 | 0.00 | 0.00 | 0.00 | 0.04 | 0.04 | 0.93 | 0.01 | 0.00 |
| 6 | 1.60 | 0.02 | 0.00 | 0.00 | 0.30 | 0.30 | 0.00 | 0.46 | 0.00 |
| 7 | 2.82 | 0.80 | 0.00 | 0.00 | 0.01 | 0.01 | 0.00 | 0.01 | 0.07 |
| 8 | 7.76 | 0.05 | 0.98 | 0.98 | 0.00 | 0.00 | 0.00 | 0.00 | 0.00 |

Coldiag2
Condition number using scaled variables = 7.76
Condition Indexes and Variance-Decomposition Proportions
Condition

**Table A37.** Multicollinearity test.

|   | Index | _cons | NO$_2$ | NO$_2{}^2$ | Age | Education | Disease | Lngdp | Gender |
|---|---|---|---|---|---|---|---|---|---|
| 1 | 1.00 | 0.06 | 0.01 | 0.07 | 0.00 | 0.00 | 0.00 | 0.00 | 0.06 |
| 2 | 1.19 | 0.01 | 0.16 | 0.01 | 0.00 | 0.05 | 0.01 | 0.18 | 0.02 |
| 3 | 1.42 | 0.00 | 0.00 | 0.00 | 0.55 | 0.26 | 0.03 | 0.01 | 0.00 |
| 4 | 1.47 | 0.01 | 0.00 | 0.00 | 0.00 | 0.10 | 0.78 | 0.03 | 0.00 |
| 5 | 1.60 | 0.02 | 0.07 | 0.00 | 0.39 | 0.44 | 0.10 | 0.01 | 0.00 |
| 6 | 1.82 | 0.01 | 0.09 | 0.23 | 0.02 | 0.01 | 0.07 | 0.45 | 0.09 |
| 7 | 2.43 | 0.00 | 0.54 | 0.38 | 0.02 | 0.05 | 0.00 | 0.31 | 0.26 |
| 8 | 3.25 | 0.90 | 0.12 | 0.32 | 0.01 | 0.08 | 0.00 | 0.01 | 0.57 |

Coldiag2
Condition number using scaled variables = 3.24
Condition Indexes and Variance-Decomposition Proportions
Condition

**Table A38.** Multicollinearity test.

|   | Index | _cons | PM10 | SO$_2$ | NO$_2$ | Age | educAtion | Disease | Lngdp | Gender |
|---|---|---|---|---|---|---|---|---|---|---|
| 1 | 1.00 | 0.00 | 0.04 | 0.04 | 0.05 | 0.00 | 0.00 | 0.00 | 0.04 | 0.00 |
| 2 | 1.21 | 0.13 | 0.00 | 0.00 | 0.00 | 0.00 | 0.01 | 0.00 | 0.00 | 0.13 |
| 3 | 1.47 | 0.00 | 0.00 | 0.03 | 0.01 | 0.16 | 0.01 | 0.36 | 0.13 | 0.00 |
| 4 | 1.59 | 0.01 | 0.00 | 0.00 | 0.00 | 0.29 | 0.62 | 0.01 | 0.00 | 0.00 |
| 5 | 1.64 | 0.01 | 0.00 | 0.00 | 0.00 | 0.48 | 0.25 | 0.25 | 0.00 | 0.00 |
| 6 | 1.86 | 0.00 | 0.04 | 0.07 | 0.01 | 0.06 | 0.03 | 0.34 | 0.41 | 0.00 |
| 7 | 2.66 | 0.00 | 0.09 | 0.05 | 0.92 | 0.00 | 0.00 | 0.00 | 0.36 | 0.00 |
| 8 | 3.16 | 0.85 | 0.00 | 0.00 | 0.00 | 0.01 | 0.07 | 0.00 | 0.00 | 0.87 |
| 9 | 3.37 | 0.00 | 0.82 | 0.80 | 0.00 | 0.00 | 0.00 | 0.03 | 0.05 | 0.00 |

Coldiag2
Condition number using scaled variables = 3.37
Condition Indexes and Variance-Decomposition Proportions
Condition

**Table A39.** Multicollinearity test.

|   | Index | _cons | PM10 | PM10$^2$ | SO$_2$ | SO$_2{}^2$ | NO$_2$ | NO$_2{}^2$ | Age | Education | Disease | Lngdp | Gender |
|---|---|---|---|---|---|---|---|---|---|---|---|---|---|
| 1 | 1.00 | 0.00 | 0.01 | 0.01 | 0.01 | 0.01 | 0.00 | 0.01 | 0.00 | 0.00 | 0.00 | 0.00 | 0.01 |
| 2 | 1.23 | 0.02 | 0.01 | 0.00 | 0.01 | 0.00 | 0.02 | 0.04 | 0.00 | 0.00 | 0.00 | 0.02 | 0.03 |
| 3 | 1.65 | 0.01 | 0.00 | 0.01 | 0.01 | 0.00 | 0.03 | 0.00 | 0.04 | 0.00 | 0.13 | 0.17 | 0.02 |
| 4 | 1.88 | 0.00 | 0.00 | 0.00 | 0.00 | 0.00 | 0.00 | 0.00 | 0.27 | 0.64 | 0.00 | 0.00 | 0.00 |
| 5 | 1.91 | 0.00 | 0.00 | 0.01 | 0.00 | 0.00 | 0.02 | 0.00 | 0.55 | 0.21 | 0.04 | 0.01 | 0.02 |
| 6 | 1.98 | 0.00 | 0.00 | 0.01 | 0.00 | 0.00 | 0.01 | 0.00 | 0.10 | 0.03 | 0.74 | 0.01 | 0.00 |
| 7 | 2.55 | 0.01 | 0.00 | 0.00 | 0.02 | 0.01 | 0.07 | 0.14 | 0.03 | 0.06 | 0.01 | 0.47 | 0.14 |
| 8 | 2.65 | 0.00 | 0.00 | 0.03 | 0.04 | 0.00 | 0.04 | 0.42 | 0.01 | 0.00 | 0.05 | 0.18 | 0.11 |
| 9 | 3.78 | 0.27 | 0.01 | 0.02 | 0.30 | 0.00 | 0.12 | 0.05 | 0.01 | 0.04 | 0.01 | 0.06 | 0.46 |
| 10 | 4.01 | 0.15 | 0.37 | 0.00 | 0.38 | 0.02 | 0.00 | 0.10 | 0.00 | 0.01 | 0.02 | 0.04 | 0.14 |
| 11 | 5.23 | 0.30 | 0.17 | 0.02 | 0.23 | 0.59 | 0.24 | 0.00 | 0.00 | 0.00 | 0.00 | 0.02 | 0.05 |
| 12 | 7.13 | 0.23 | 0.44 | 0.90 | 0.01 | 0.36 | 0.45 | 0.23 | 0.00 | 0.00 | 0.00 | 0.00 | 0.01 |

Coldiag2
Condition number using scaled variables = 7.13
Condition Indexes and Variance-Decomposition Proportions
Condition

*Appendix C.2. Multicollinearity Test for OLS Regression*

We used VIF to test the multicollinearity problems. estat vif calculates the centered or uncentered variance inflation factors (VIFs) for the independent variables specified in a linear regression model.

**Table A40.** Multicollinearity test.

| Variable | VIF | 1/VIF |
|----------|-----|-------|
| Education | 1.08 | 0.928736 |
| Health | 1.06 | 0.939754 |
| Lngdp | 1.05 | 0.951208 |
| Gender | 1.05 | 0.952480 |
| Disease | 1.04 | 0.962520 |
| age | 1.02 | 0.979863 |
| Mean VIF | 1.05 | |

**Table A41.** Multicollinearity test.

| Variable | VIF | 1/VIF |
|----------|-----|-------|
| $NO_2$ | 1.76 | 0.568128 |
| PM10 | 1.53 | 0.654072 |
| Lngdp | 1.30 | 0.771432 |
| Education | 1.08 | 0.927595 |
| Health | 1.07 | 0.931172 |
| Disease | 1.06 | 0.942868 |
| Gender | 1.05 | 0.952106 |
| Age | 1.02 | 0.978021 |
| $SO_2$ | 1.01 | 0.991937 |
| Mean VIF | 1.21 | |

**Table A42.** Multicollinearity test.

| Variable | VIF | 1/VIF |
|----------|-----|-------|
| $NO_2$ * health | 7.81 | 0.127996 |
| health | 4.83 | 0.207184 |
| PM10 * health | 4.39 | 0.227780 |
| Lngdp | 1.19 | 0.839801 |
| Education | 1.07 | 0.931340 |
| Disease | 1.06 | 0.944869 |
| Gender | 1.05 | 0.954871 |
| Age | 1.02 | 0.980083 |
| $SO_2$&health | 1.01 | 0.989930 |
| Mean VIF | 2.60 | |

**Table A43.** Multicollinearity test.

| Variable | VIF | 1/VIF |
|----------|-----|-------|
| $NO_2$ * health | 7.81 | 0.127996 |
| health | 4.83 | 0.207184 |
| PM10 * health | 4.39 | 0.227780 |
| Lngdp | 1.19 | 0.839801 |
| Education | 1.07 | 0.931340 |
| Disease | 1.06 | 0.944869 |
| Gender | 1.05 | 0.954871 |
| Age | 1.02 | 0.980083 |
| $SO_2$&health | 1.01 | 0.989930 |
| Mean VIF | 2.60 | |

**Table A44.** Multicollinearity test.

| Variable | VIF | 1/VIF |
|:---:|:---:|:---:|
| PM10 | 1.86 | 0.537684 |
| $PM10^2$ | 1.86 | 0.537684 |
| Mean VIF | 1.86 | |

**Table A45.** Multicollinearity test.

| Variable | VIF | 1/VIF |
|:---:|:---:|:---:|
| $SO_2$ | 15.43 | 0.064801 |
| $SO_2^2$ | 15.43 | 0.064801 |
| Mean VIF | 15.43 | |

**Table A46.** Multicollinearity test.

| Variable | VIF | 1/VIF |
|:---:|:---:|:---:|
| $NO_2$ | 1.04 | 0.960136 |
| $NO_2^2$ | 1.04 | 0.960136 |
| Mean VIF | 1.04 | |

**Table A47.** Multicollinearity test.

| Variable | VIF | 1/VIF |
|:---:|:---:|:---:|
| $NO_2$ | 1.63 | 0.615186 |
| PM10 | 1.60 | 0.626726 |
| $SO_2$ | 1.03 | 0.970426 |
| Mean VIF | 1.42 | |

**Table A48.** Multicollinearity test.

| Variable | VIF | 1/VIF |
|:---:|:---:|:---:|
| $NO_2^2$ | 1.17 | 0.856324 |
| $NO_2$ | 1.83 | 0.545422 |
| $SO_2$ | 16.05 | 0.062292 |
| $SO_2^2$ | 15.70 | 0.063707 |
| PM10 | 3.17 | 0.315347 |
| $PM10^2$ | 2.15 | 0.466159 |
| Mean VIF | 1.42 | |

**Table A49.** Multicollinearity test.

| Variable | VIF | 1/VIF |
|:---:|:---:|:---:|
| PM10 | 2.18 | 0.459763 |
| $PM10^2$ | 1.99 | 0.502957 |
| Lngdp | 1.19 | 0.837080 |
| Education | 1.07 | 0.930635 |
| Gender | 1.05 | 0.951440 |
| Age | 1.02 | 0.979703 |
| disease | 1.01 | 0.988049 |
| Mean VIF | 1.36 | |

**Table A50.** Multicollinearity test.

| Variable | VIF | 1/VIF |
|---|---|---|
| $SO_2$ | 15.47 | 0.064639 |
| $SO_2{}^2$ | 15.45 | 0.064723 |
| Education | 1.07 | 0.930403 |
| Gender | 1.05 | 0.950717 |
| Lngsp | 1.03 | 0.969817 |
| Age | 1.02 | 0.979583 |
| disease | 1.00 | 0.996117 |
| Mean VIF | 5.16 | |

**Table A51.** Multicollinearity test.

| Variable | VIF | 1/VIF |
|---|---|---|
| $NO_2$ | 27.57 | 0.036275 |
| $NO_2{}^2$ | 26.99 | 0.037055 |
| Lngdp | 1.30 | 0.769348 |
| Education | 1.08 | 0.927188 |
| Gender | 1.05 | 0.950564 |
| Age | 1.02 | 0.978403 |
| disease | 1.01 | 0.986874 |
| Mean VIF | 8.57 | |

**Table A52.** Multicollinearity test.

| Variable | VIF | 1/VIF |
|---|---|---|
| $NO_2$ | 1.33 | 0.752278 |
| Lngdp | 1.30 | 0.769348 |
| Education | 1.08 | 0.927188 |
| $NO_2{}^2$ | 1.05 | 0.947888 |
| Gender | 1.05 | 0.950564 |
| Age | 1.02 | 0.978403 |
| disease | 1.01 | 0.986874 |
| Mean VIF | 1.12 | |

**Table A53.** Multicollinearity test.

| Variable | VIF | 1/VIF |
|---|---|---|
| $NO_2$ | 1.88 | 0.532724 |
| PM10 | 1.60 | 0.624213 |
| Lngdp | 1.29 | 0.772899 |
| Education | 1.07 | 0.930323 |
| Gender | 1.05 | 0.951390 |
| $SO_2$ | 1.03 | 0.967910 |
| age | 1.02 | 0.978653 |
| disease | 1.01 | 0.987308 |
| Mean VIF | 1.25 | |

**Table A54.** Multicollinearity test.

| Variable | VIF | 1/VIF |
|---|---|---|
| PM10 | 1.86 | 0.537684 |
| $PM10^2$ | 1.86 | 0.537684 |
| Mean VIF | 1.86 | |

**Table A55.** Multicollinearity test.

| Variable | VIF | 1/VIF |
|:--------:|:---:|:-----:|
| $SO_2$ | 15.43 | 0.064801 |
| $SO_2^2$ | 15.43 | 0.064801 |
| Mean VIF | 15.43 | |

**Table A56.** Multicollinearity test.

| Variable | VIF | 1/VIF |
|:--------:|:---:|:-----:|
| $NO_2$ | 1.04 | 0.960136 |
| $NO_2^2$ | 1.04 | 0.960136 |
| Mean VIF | 1.04 | |

**Table A57.** Multicollinearity test.

| Variable | VIF | 1/VIF |
|:--------:|:---:|:-----:|
| $NO_2$ | 1.63 | 0.615186 |
| PM10 | 1.60 | 0.626726 |
| $SO_2$ | 1.03 | 0.970426 |
| Mean VIF | 1.42 | |

**Table A58.** Multicollinearity test.

| Variable | VIF | 1/VIF |
|:--------:|:---:|:-----:|
| $SO_2$ | 16.05 | 0.062292 |
| $SO_2^2$ | 15.70 | 0.063707 |
| PM10 | 3.17 | 0.315347 |
| $PM10^2$ | 2.15 | 0.466159 |
| $NO_2$ | 1.83 | 0.545422 |
| $NO_2^2$ | 1.17 | 0.856324 |
| Mean VIF | 6.68 | |

**Table A59.** Multicollinearity test.

| Variable | VIF | 1/VIF |
|:--------:|:---:|:-----:|
| PM10 | 2.18 | 0.459763 |
| $PM10^2$ | 1.99 | 0.502957 |
| Lngdp | 1.19 | 0.837080 |
| Education | 1.07 | 0.930635 |
| Gender | 1.05 | 0.951440 |
| Age | 1.02 | 0.979703 |
| disease | 1.01 | 0.988049 |
| Mean VIF | 1.36 | |

**Table A60.** Multicollinearity test.

| Variable | VIF | 1/VIF |
|:--------:|:---:|:-----:|
| $SO_2$ | 15.47 | 0.064639 |
| $SO_2^2$ | 15.45 | 0.064723 |
| Education | 1.07 | 0.930403 |
| Gender | 1.05 | 0.950717 |
| Lngsp | 1.03 | 0.969817 |
| Age | 1.02 | 0.979583 |
| disease | 1.00 | 0.996117 |
| Mean VIF | 5.16 | |

**Table A61.** Multicollinearity test.

| Variable | VIF | 1/VIF |
|---|---|---|
| $NO_2$ | 1.33 | 0.752270 |
| Lngdp | 1.30 | 0.769348 |
| Education | 1.08 | 0.927188 |
| $NO_2$ | 1.05 | 0.947888 |
| Gender | 1.05 | 0.950564 |
| Age | 1.02 | 0.978403 |
| disease | 1.01 | 0.986874 |
| Mean VIF | 1.12 | |

**Table A62.** Multicollinearity test.

| Variable | VIF | 1/VIF |
|---|---|---|
| $NO_2$ | 1.88 | 0.532724 |
| PM10 | 1.60 | 0.624213 |
| Lngdp | 1.29 | 0.772899 |
| Education | 1.07 | 0.930323 |
| Gender | 1.05 | 0.951390 |
| $SO_2$ | 1.03 | 0.967910 |
| age | 1.02 | 0.978635 |
| disease | 1.01 | 0.987308 |
| Mean VIF | 1.25 | |

**Table A63.** Multicollinearity test.

| Variable | VIF | 1/VIF |
|---|---|---|
| $SO_2$ | 16.07 | 0.062228 |
| $SO_2^2$ | 15.71 | 0.063668 |
| PM10 | 3.22 | 0.310308 |
| $PM10^2$ | 2.19 | 0.455695 |
| $NO_2$ | 2.04 | 0.491125 |
| Lngdp | 1.32 | 0.755907 |
| $NO_2$ | 1.18 | 0.849807 |
| education | 1.08 | 0.926813 |
| Gender | 1.05 | 0.951080 |
| Age | 1.02 | 0.978368 |
| disease | 1.02 | 0.981867 |
| Mean VIF | 4.17 | |

## Appendix D. Residential Analysis

For OLS regression models, we used the residential test for each regression. The final results are shown below.

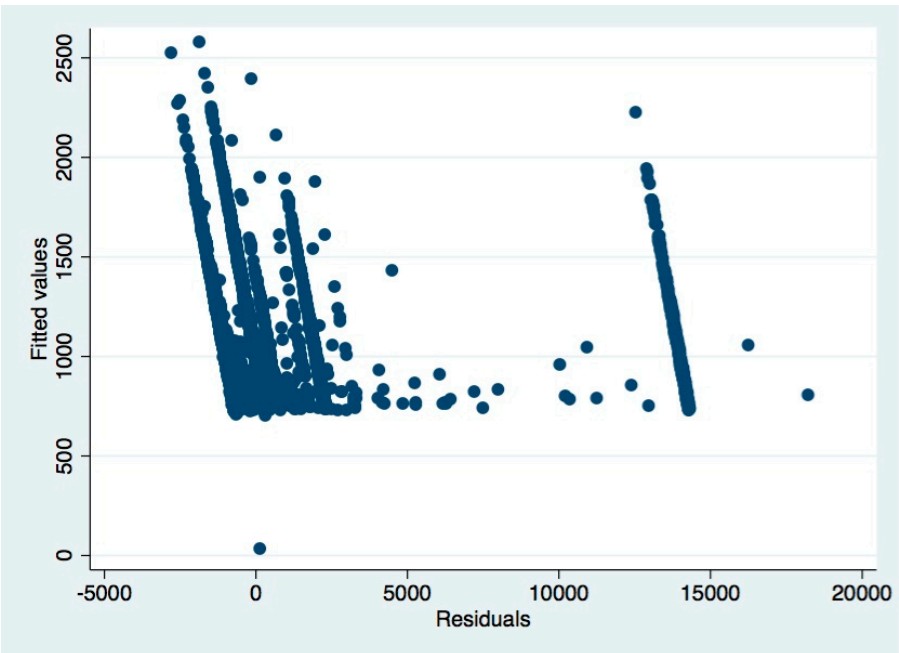

**Figure A2.** Residential test.

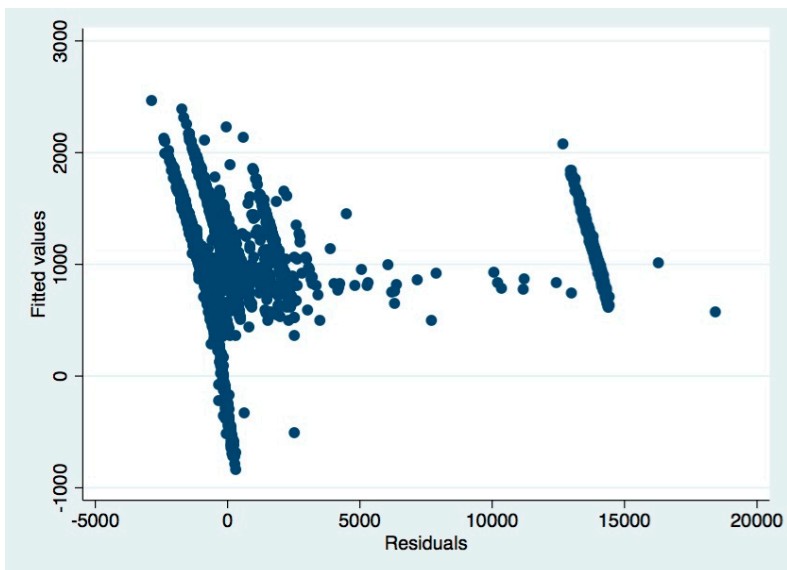

**Figure A3.** Residential test.

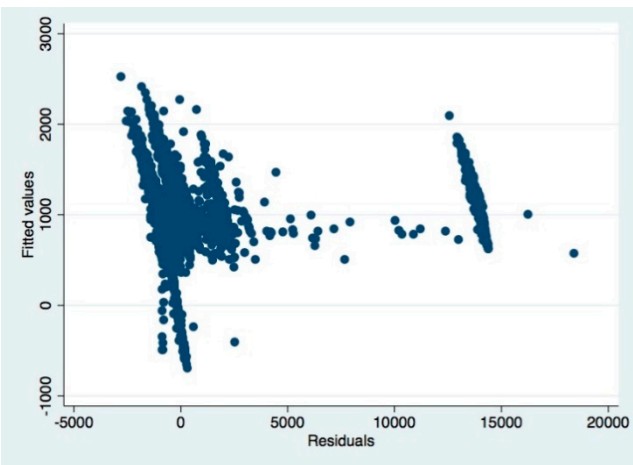

**Figure A4.** Residential test.

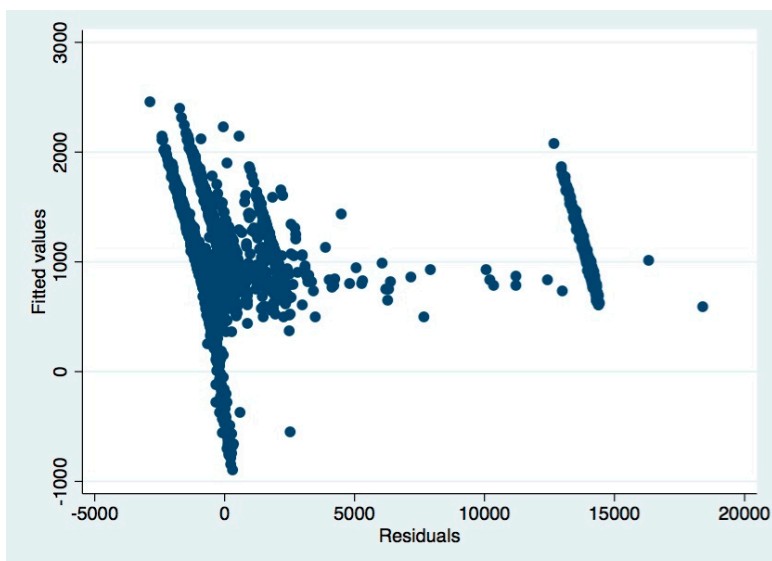

**Figure A5.** Residential test.

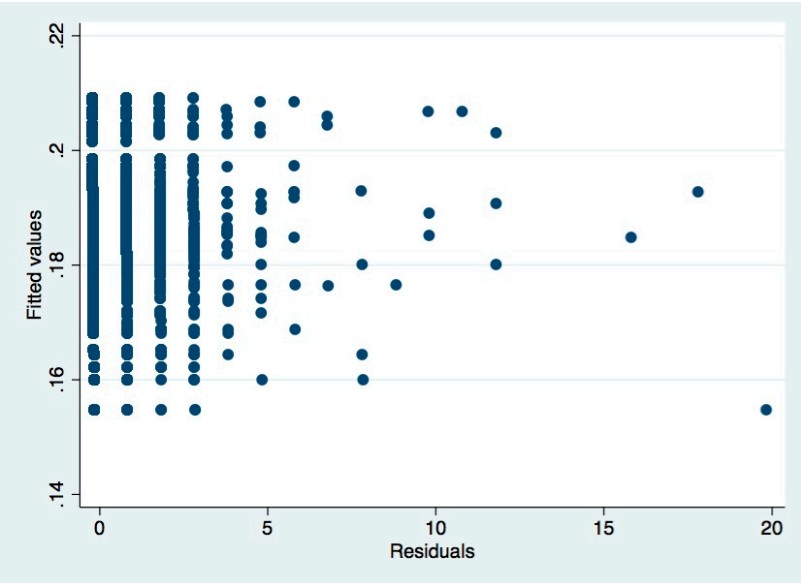

**Figure A6.** Residential test.

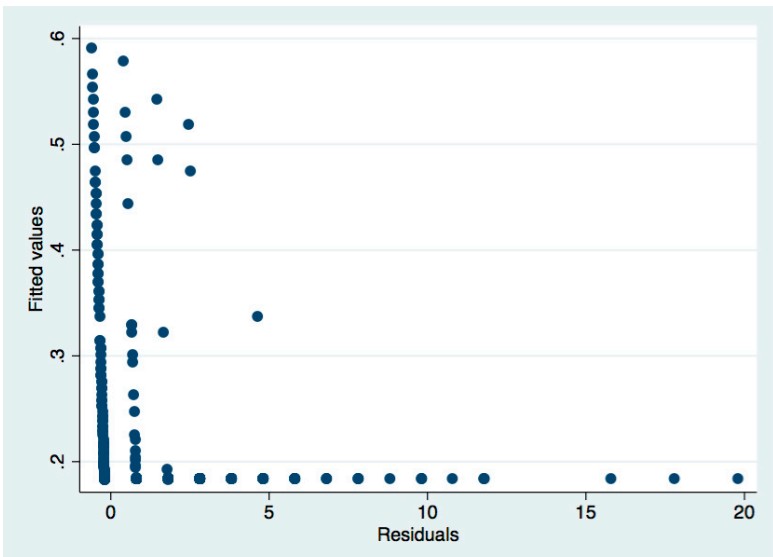

**Figure A7.** Residential test.

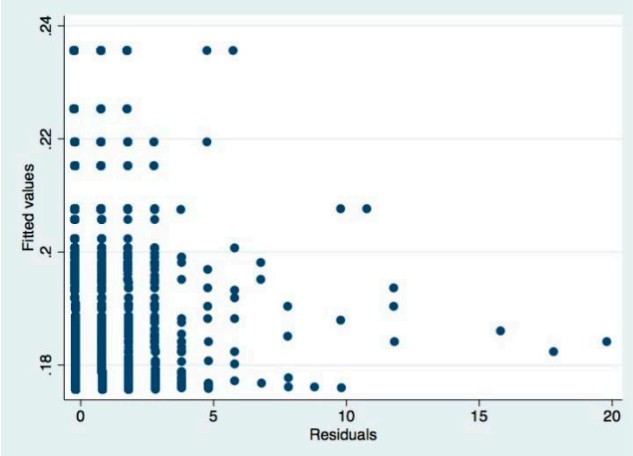

**Figure A8.** Residential test.

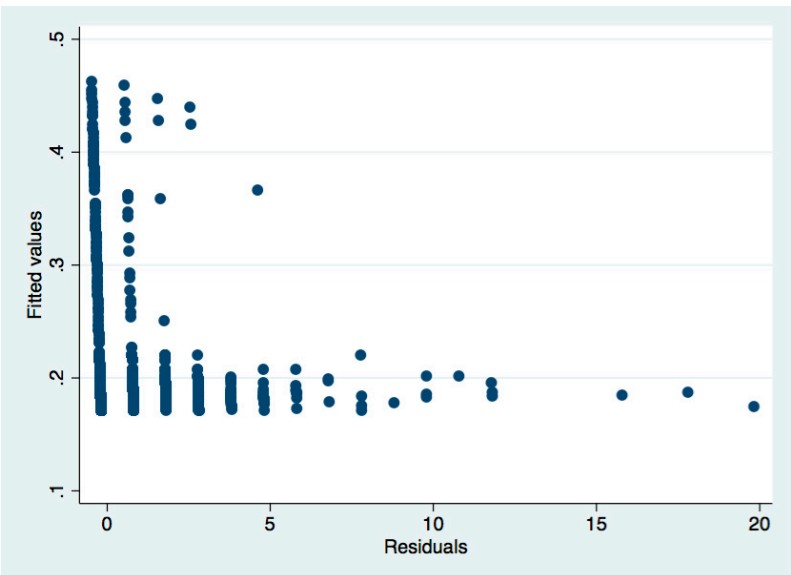

**Figure A9.** Residential test.

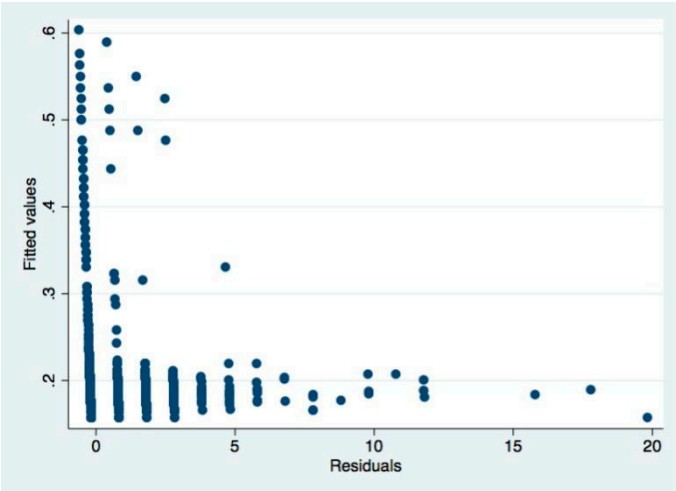

**Figure A10.** Residential test.

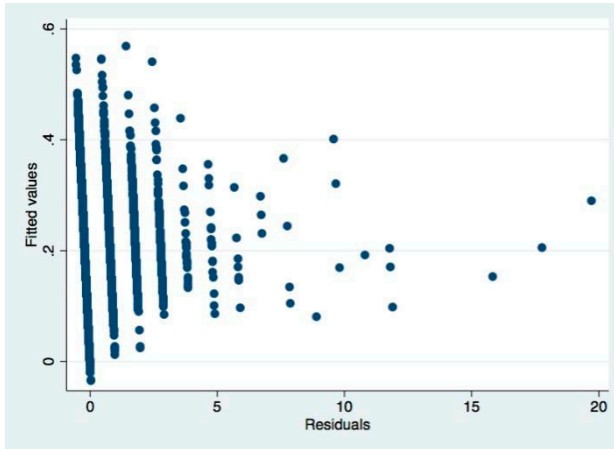

**Figure A11.** Residential test.

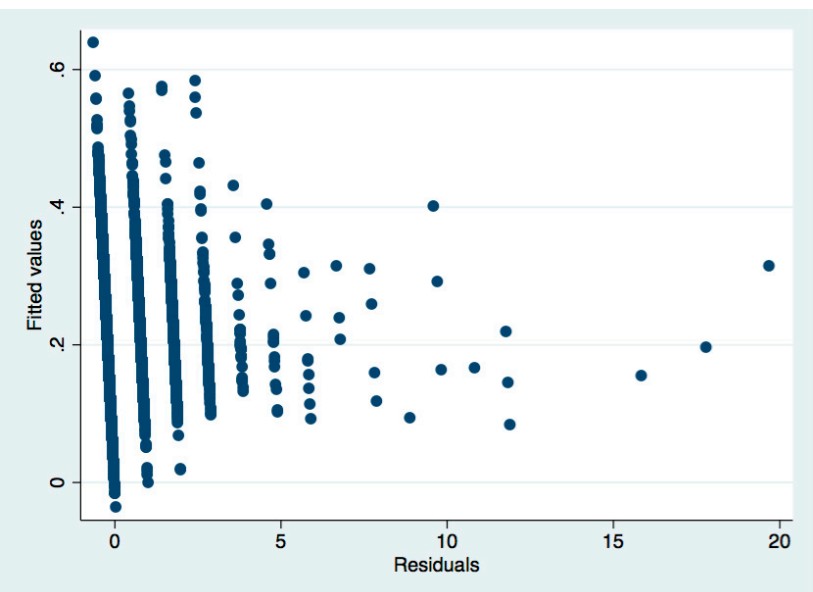

**Figure A12.** Residential test.

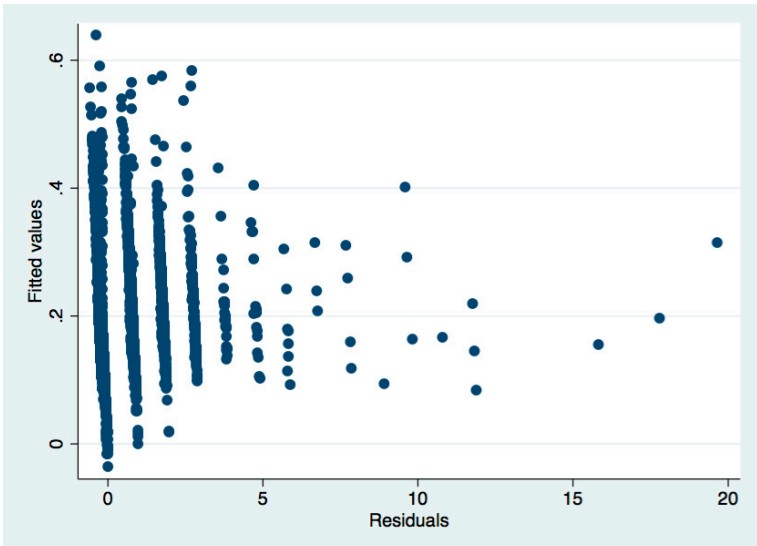

**Figure A13.** Residential test.

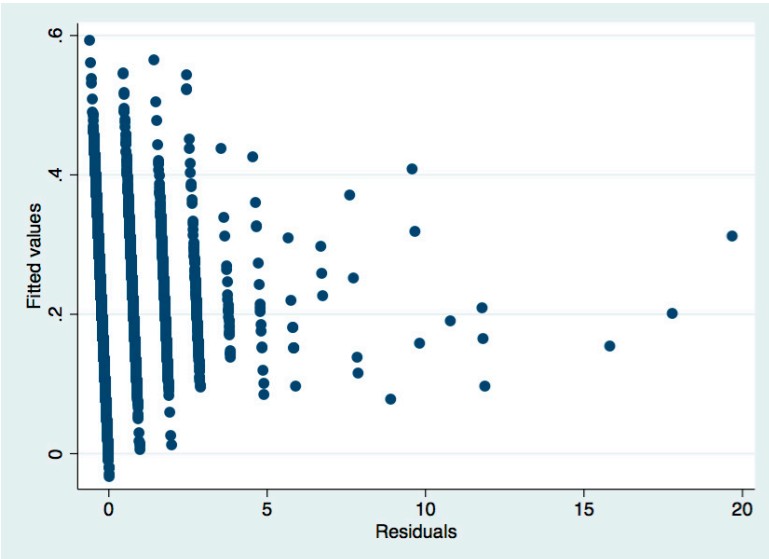

**Figure A14.** Residential test.

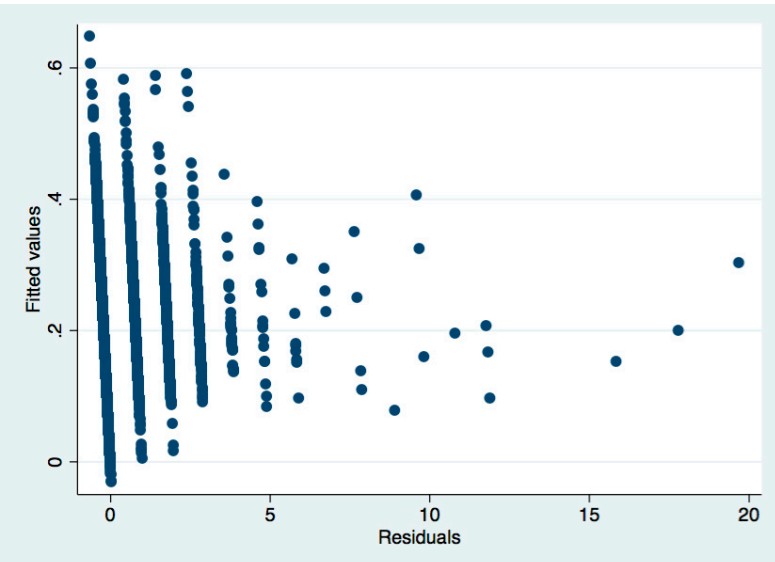

**Figure A15.** Residential test.

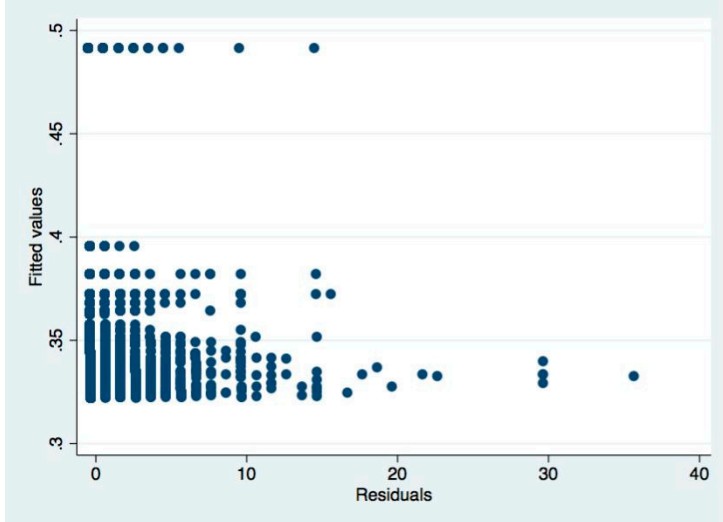

**Figure A16.** Residential test.

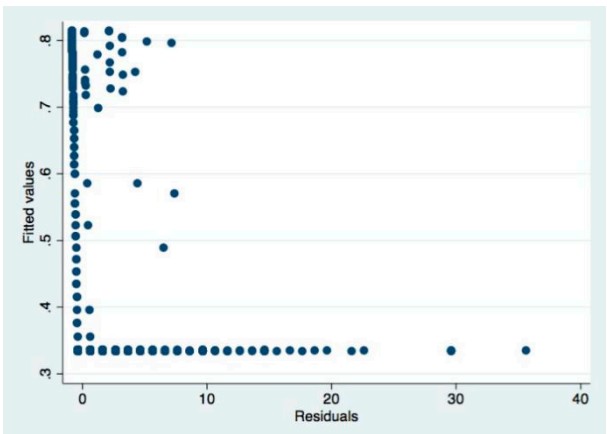

**Figure A17.** Residential test.

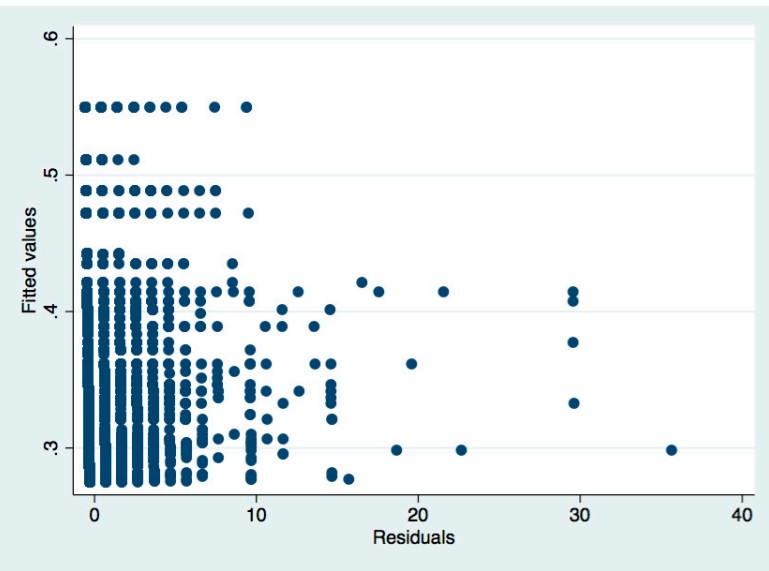

**Figure A18.** Residential test.

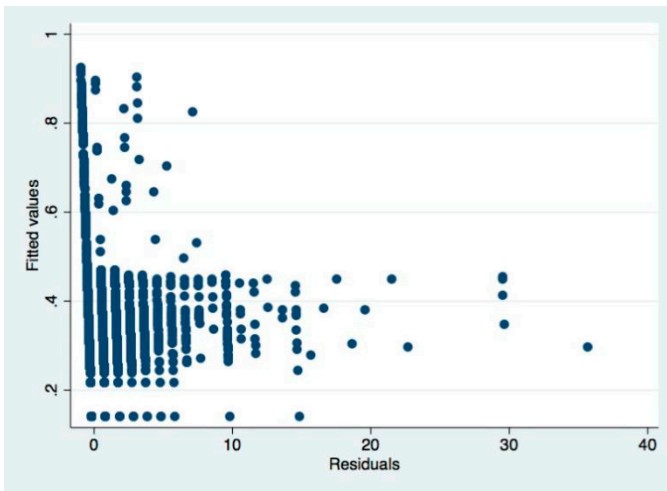

**Figure A19.** Residential test.

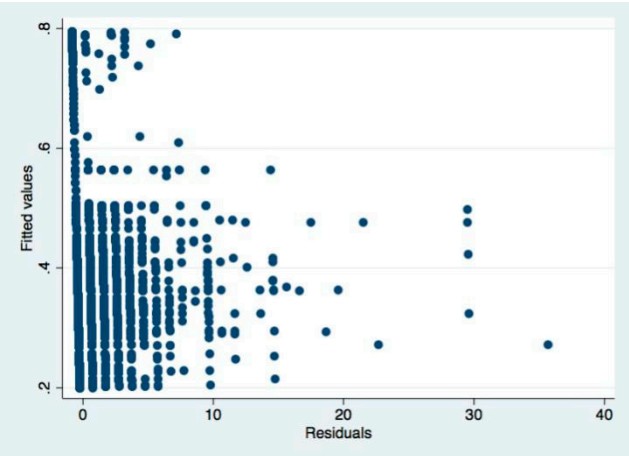

**Figure A20.** Residential test.

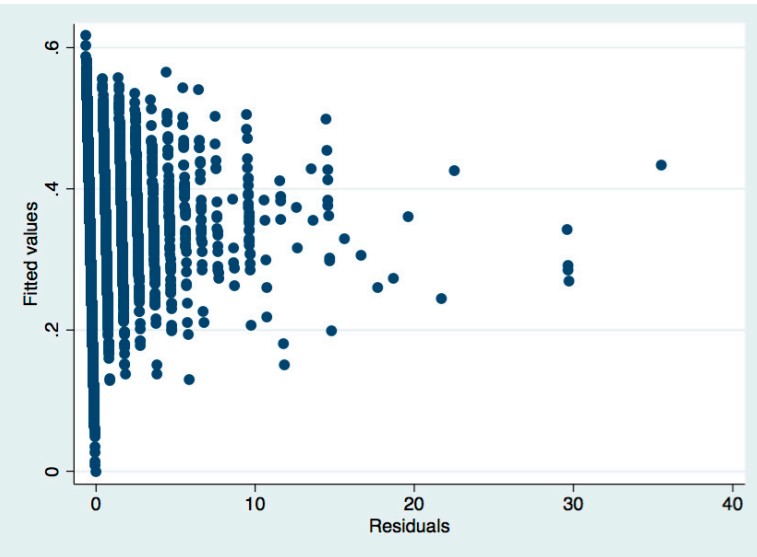

**Figure A21.** Residential test.

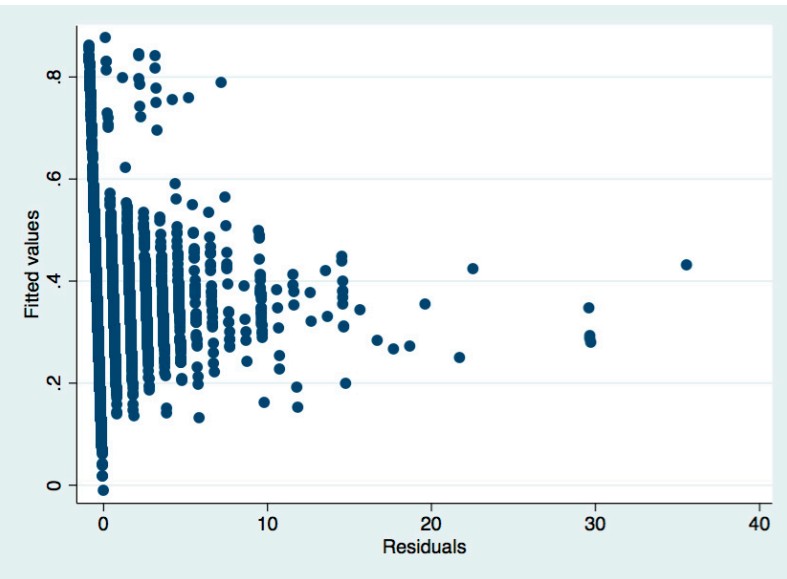

**Figure A22.** Residential test.

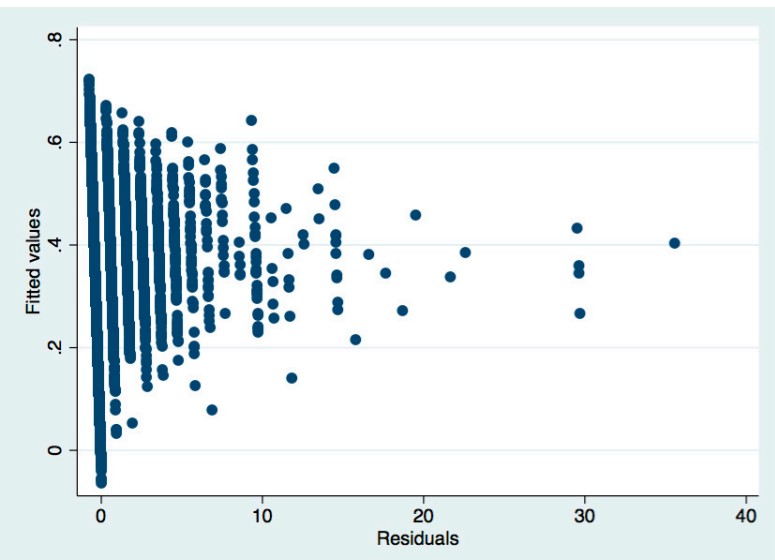

**Figure A23.** Residential test.

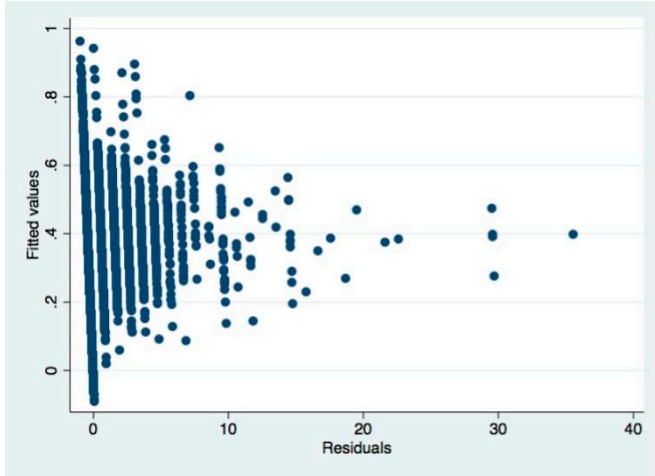

**Figure A24.** Residential test.

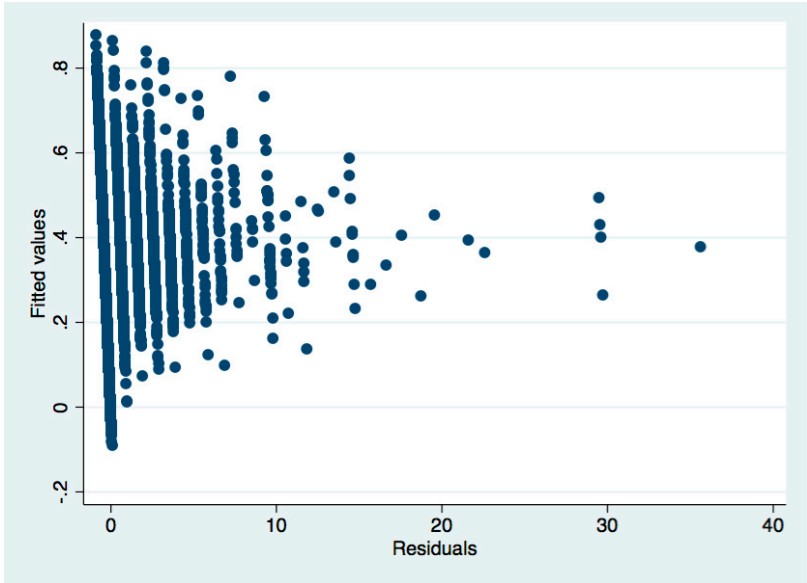

**Figure A25.** Residential test.

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
