# Peer review of "Does Air Pollution Affect Health and Medical Insurance Cost in the Elderly: An Empirical Evidence from China"

_sustainability, doi:10.3390/su11061526_

Reviewer 1 Report

see attachment

Author Response

Thank you very much for your letter dated 12 February 2019 and the review reports. Based on your comment and request, we have made extensive modification on the original manuscript. We attached revised manuscript in the formats of both PDF and MS word for your approval. A document answering questions from the referees was also summarized and enclosed. A revised manuscript with the correction sections red marked was attached as the supplemental material for easy check. Should you have questions, please contact us without hesitation.Please check out the attachment for more details.

Yours sincerely,

Tianlei Pi

E-mail: pitianlei@cqu.edu.cn

Reviewer 2 Report

Please refer to the attached report.

Author Response

Dear Editors and Reviewers:

Thank you very much for your letter dated 12 February 2019 and the review reports. Based on your comment and request, we have made extensive modification on the original manuscript. We attached revised manuscript in the formats of both PDF and MS word for your approval. A document answering questions from the referees was also summarized and enclosed. A revised manuscript with the correction sections red marked was attached as the supplemental material for easy check. Should you have questions, please contact us without hesitation. Please check out the attachment for more details.

Yours sincerely,

Tianlei Pi

E-mail: pitianlei@cqu.edu.cn

Round  2

Reviewer 1 Report

The manuscript has improved. Based on the new manuscript and the letters to reviewers, I have a few minor comments:

Please make sure that you have uploaded the latest manuscript. According to the letter to reviewers, you have run additional tests and present the results in Appendices. In the pdf-manuscript, I cannot see newly added Appendices. In addition, there seems to be a mismatch in line numbers between the manuscript and what’s referenced in the letters to reviewers.

The citation style is inconsistent and very confusing. Sometimes you use “Author (year)”, sometimes “(number)”. You also have the same numbering style for models and equations, which adds to the confusion. Please correct that at final proofing.

In Table 2, Column “Unit” Row 4, “Dumb” is still not corrected (should be “Dummy”).

Author Response

Thank you very much for your letter and review report dated on 25 February 2019. Based on your comments and suggestions, we have made a further modify on the original manuscript. Here, we attached revised manuscript in the formats of both PDF and MS word for your convenience. A document answering every question from the referees was also summarized and enclosed. A revised manuscript with the correction sections red marked was attached as the supplemental material and for easy checking or editing purpose. Should you have questions, please contact us without any hesitation. Please check out the attachment for more details.

Yours sincerely,

Tianlei Pi

E-mail: pitianlei@cqu.edu.c

Reviewer 2 Report

I would like to thank the authors for taking all the comments into account and for providing detailed answers.

There are some inconsistencies in the revised manuscript that, in my opinion, should be corrected to make the paper publishable:

- the results in Tables 6, 7, 9, 10, 12 were modified. In the new version, many standard errors appear to be considerably larger than parameter estimates (which are nevertheless marked with asterisks) and some standard errors have negative sign. Please check these tables for possible errors/typos;

- in your response file, you display VIF statistics on p. 19 for a model without quadratic terms. Please compute this statistic in relation to quadratic terms, as those are likely to induce multicollinearity, and report the results in the main body of the article;

- in panel regressions (reported in your response file), residual plots and results of the White test for heteroscedasticity do not seem to support the OLS hypotheses. What about pooled regressions reported in the main body of the paper? Could you please include residual analysis and White test for these models in the main body of the paper and comment on results?

- the meaning of "microcosmic samples" is unclear to me. Do you mean sample units?

Thank you for your consideration. Best regards. 

Author Response

Thank you very much for your letter and review report dated on 25 February 2019. Based on your comments and suggestions, we have made a further modify on the original manuscript. Here, we attached revised manuscript in the formats of both PDF and MS word for your convenience. A document answering every question from the referees was also summarized and enclosed. A revised manuscript with the correction sections red marked was attached as the supplemental material and for easy checking or editing purpose. Should you have questions, please contact us without any hesitation. Please check out the attachment for more details.

Yours sincerely,

Tianlei Pi

E-mail: pitianlei@cqu.edu.cn
